# The *ABCG2* Q141K hyperuricemia and gout associated variant illuminates the physiology of human urate excretion

Kazi Mirajul Hoque[1], Eryn E. Dixon [1], Raychel M. Lewis[1], Jordyn Allan[2], Gregory D. Gamble[2],
Amanda J. Phipps-Green [3], Victoria L. Halperin Kuhns [1], Anne M. Horne[2], Lisa K. Stamp[4],
Tony R. Merriman [3], Nicola Dalbeth [2] & Owen M. Woodward [1]✉

The pathophysiological nature of the common *ABCG2* gout and hyperuricemia associated variant Q141K (rs2231142) remains undefined. Here, we use a human interventional cohort study (ACTRN12615001302549) to understand the physiological role of ABCG2 and find that participants with the Q141K *ABCG2* variant display elevated serum urate, unaltered FEUA, and significant evidence of reduced extra-renal urate excretion. We explore mechanisms by generating a mouse model of the orthologous Q140K *Abcg2* variant and find male mice have significant hyperuricemia and metabolic alterations, but only subtle alterations of renal urate excretion and ABCG2 abundance. By contrast, these mice display a severe defect in ABCG2 abundance and function in the intestinal tract. These results suggest a tissue specific pathobiology of the Q141K variant, support an important role for ABCG2 in urate excretion in both the human kidney and intestinal tract, and provide insight into the importance of intestinal urate excretion for serum urate homeostasis.

[1] Department of Physiology, University of Maryland School of Medicine, Baltimore, MD, USA. [2] Department of Medicine, University of Auckland, Auckland, New Zealand. [3] Department of Biochemistry, University of Otago, Dunedin, New Zealand. [4] Department of Medicine, University of Otago, Christchurch, New Zealand. ✉email: owoodward@som.umaryland.edu

Uric acid is the terminal metabolite of purine metabolism in humans as a result of the pseudogenization of the uricase (*UOX*) gene[1]. Increased levels (>7 mg/dL) of serum urate (predominant form at physiological pH) is defined as hyperuricemia and is present in 43 million Americans[1]. Hyperuricemia may lead to saturation and precipitation of the weakly soluble urate as monosodium urate crystals, occasionally in the renal tubules causing kidney stones, but most often in the synovial fluid of joints, causing gout, the most common form of inflammatory arthritis[2]. The heritability of hyperuricemia is high, 70%[3,4], predicting important genetic components to hyperuricemia and subsequent risk of gout. In observational studies, hyperuricemia increases risk of hypertension incidence[5], and asymptomatic hyperuricemia in the absence of comorbidities increases the risk of hypertension, chronic kidney disease, and obesity 2–3 fold[6]. However, demonstrating causality of hyperuricemia in the pathophysiology of hypertension or metabolic diseases has been complicated by the reciprocal relationship between target organs for these disorders, the kidney and liver respectively, and the function of these organs in urate production and excretion. Further, we lack mechanistic understanding in how urate excretion is mediated and how urate homeostasis is regulated, even as important pieces to these puzzles have been recently identified[7,8].

Humans excrete urate through the gastrointestinal tract and the kidneys, where urate is freely filtered at the glomerulus, mostly reabsorbed in the early proximal tubule (S1), and secreted and possibly reabsorbed in the S2–S3 segments of the proximal tubule[1], although this remains controversial[9]. Genetic approaches identified a number of kidney and gastrointestinal tract transporters with significant affinity for urate. These include the SLC transporter genes, *SLC22A12* (refs. [10,11]), *SLC2A9* (refs. [11,12]), and the ABC transporter gene, *ABCG2* (ref.[13]). Variants in these three genes alone contribute 5% of measured variability in serum urate (SU), significantly more than all other variants combined[14]. Functional studies in humans and model systems have demonstrated that ABCG2 (refs.[15,16]) and SLC2A9/GLUT9 (ref.[17]) have a role in intestinal excretion, although specific cellular mechanisms remain undescribed. In the kidney, URAT1 (*SLC22A12*) has proven an important conduit for urate reabsorption[18], but the functional roles, expression, localization, and regulation of SLC2A9 and ABCG2 remains poorly understood.

The *ABCG2* and *SLC2A9* (ref.[19]) loci harbor common human single nucleotide polymorphisms (SNPs) that associate with increased serum urate levels, including rs2231142, resulting in a missense variation in the ABCG2 protein, p.Gln141Lys (Q141K), found in hundreds of millions of individuals[1,20]. Interestingly, *ABCG2* polymorphisms appear to confer gout risk through pleiotropic pathways, contributing both in the presence of hyperuricemia, and independent of increases in serum urate[21]. One of the most puzzling aspects of understanding the Q141K *ABCG2* variant is how and where it affects urate excretion. In vitro, the Q141K protein is a partial loss of function protein[13,22] with increased instability and frequency of degradation, resulting in both function and significant abundance defects[23]. Mouse models of *ABCG2* knockout show missing urate transport in both the intestines and the kidney[16], but studies of humans with the Q141K variant have been less consistent. Previous large association studies have reported significant increases[24], significant decreases[25], or no alterations at all[26] in renal fractional excretion of urate for individuals possessing the minor allele (T, corresponding to 141K) of rs2231142. These inconclusive studies have led to doubt for the role of ABCG2 in renal excretion of urate. Interestingly, RNA-seq has found ABCG2 mRNA in the human kidney[27,28] and previous studies have documented protein expression and transport function in the apical brush border of renal

epithelia[29]. Fully understanding the pathological role of the Q141K *ABCG2* variant allele in urate handling is important for increasing our understanding of the pathogenic nature of urate.

Here, we use a human interventional study and a CRISPR knock-in mouse model of the orthologous Q140K *Abcg2* to better understand the role of ABCG2 in urate excretion. Our results support the use of the mouse as a model for ABCG2-mediated urate handling in humans, support the role of ABCG2 in both renal and intestinal excretion of urate, and illuminates the complexity of normal and pathological urate excretion.

## Results

**Human interventional renal urate handling study.** To understand the impact of ABCG2 and its common variant (*ABCG2*: rs2231142, Q141K) on renal urate handling, we provoked acute increases in serum urate by oral administration of the purine nucleoside, inosine, to 100 healthy human volunteers. Inosine is quickly metabolized into urate (see "Methods" section) and allows analysis of the effects of a standardized purine load on both SU concentrations and renal uric acid handling. Participants were young, had no history of gout or being treated with urate-lowering therapies (population characteristics, Supplementary Table 1), and were recruited from two specific populations—people of Polynesian ancestry (New Zealand Māori and Pacific peoples) and people of European ancestry. Oral administration of inosine caused a rapid increase in SU levels in all individuals regardless of *ABCG2* genotype (Fig. 1a), and the rates of increases were similar for all participants (ANCOVA, $p_{time} < 0.0001$) suggesting that *ABCG2* genotypes do not have broad effects on the absorption of inosine nor the metabolism of inosine into urate. The absolute levels of SU differed significantly, however, among the *ABCG2* genotypes, both at baseline and at saturation (Fig. 1a), with individuals possessing at least one copy of the *ABCG2* risk allele (141K) having significantly elevated SU at baseline and throughout (ANCOVA, $p_{time} < 0.0001$, $p_{SNP} = 0.0007$) as compared to individuals without the risk allele (Q141). The maintenance of the relative SU differences in individuals with and without 141K across a range of SU levels suggests no energetic homeostatic mechanisms are activated in the time course of the experiment dependent on or affected by *ABCG2* genotype. A secondary analysis using a stratification by ancestry showed similar results (Supplementary Fig. 1, see "Methods" section).

The kidney is responsible for the majority of urate excretion in humans, and renal excretion is dependent on filtered urate load (GFR*SU) and the fraction of the filtered load (FEUA) that ultimately ends up in the excreted urine[9]. A comparison of FEUA in participants with (141K) and without (Q141) the *ABCG2* risk variant revealed no differences in mean FEUA (Fig. 1b). This is in contrast to the individuals with the previously described urate-lowering associated variant allele of *SLC2A9* rs11942223 (ref.[30]), which exhibited a significant alteration in the renal excretion of urate, consistent with the hypothesized role of SLC2A9 in renal reabsorption (Supplementary Fig. 2). We also measured urinary urate excretion (UUE, calculated by normalizing urinary urate to urinary creatinine) and found UUE increased through time and with urate load, but there were no differences between the Q141 and 141K individuals (Fig. 1c). Together, these data suggest the kidney was passively spilling the increased filtered load, not using active processes to adjust to changing loads.

Next, we looked closer at the relationship between kidney handling of urate, as assessed by FEUA, and observed variation in SU[18]. First, we found that FEUA and SU were significantly correlated (Supplementary Fig. 1b) at all time points. Second, we found that SU level was dependent on FEUA for both *ABCG2* genotypes, at all time points (Fig. 1c, d, linear regression,

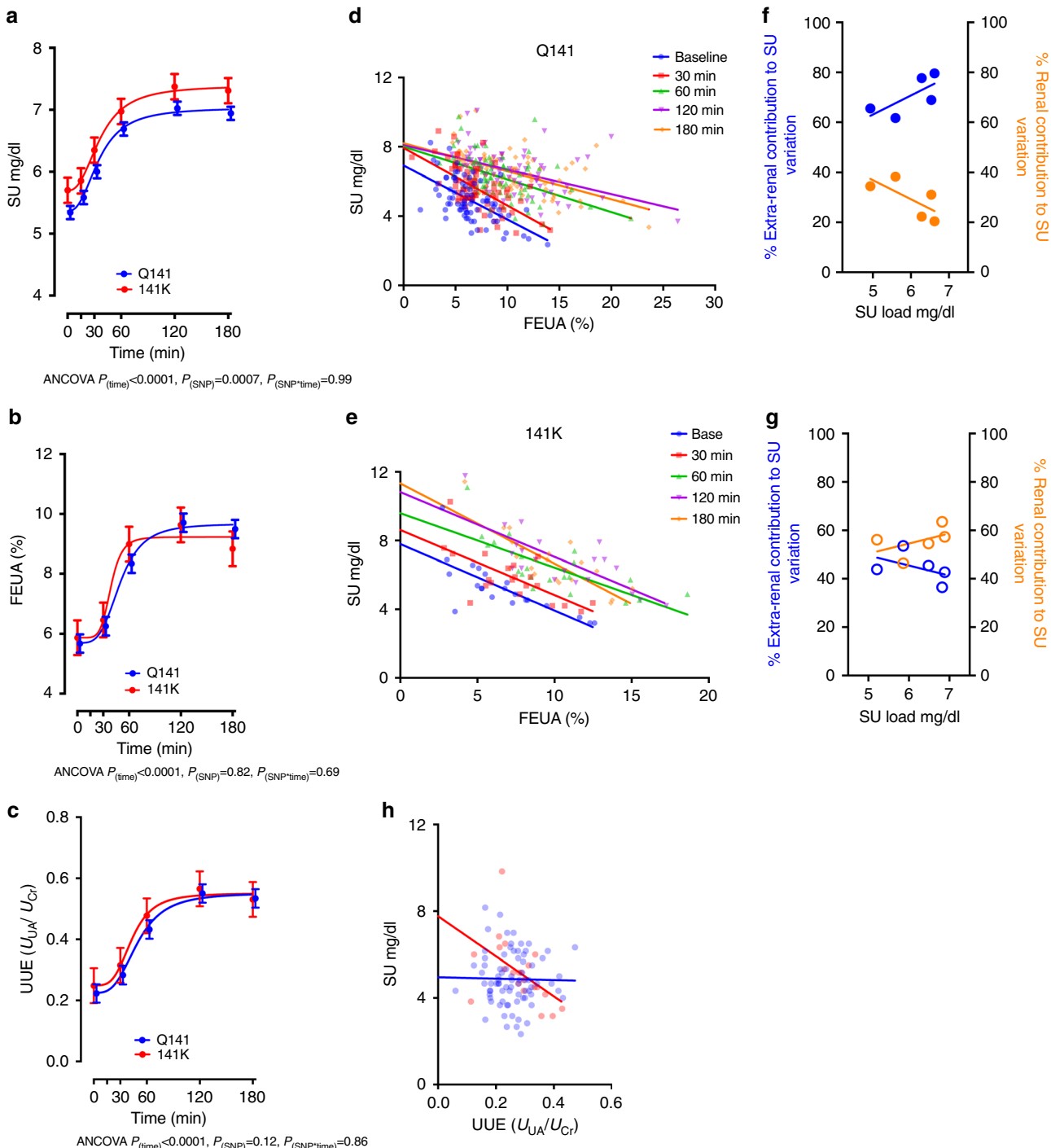

**Fig. 1 Human interventional study demonstrates significant alterations in urate handling in individuals with the 141K allele.** Effect of ABCG2 genotypes (Q141, $n = 79$ and 141K, $n = 21$ participants, Supplementary Table 1 for characteristics of participants) on **a** serum urate ($p_{(time)} < 0.0001$; $p_{(SNP)} = 0.0007$; $p_{(SNP*time)} = 0.99$), **b** fractional excretion of urate (FEUA)($p_{(time)} < 0.0001$; $p_{(SNP)} = 0.82$; $p_{(SNP*time)} = 0.69$), and **c** urinary excretion (UUE $= U_{UA}/U_{Cr}$) ($p_{(time)} < 0.0001$; $p_{(SNP)} = 0.12$; $p_{(SNP*time)} = 0.86$) following inosine load to entire group (**a–c**, statistical analysis: two-tailed ANCOVAs adjusted for age, sex, BMI, and ancestry; ± standard error of the mean [SEM]). Effect of ABCG2 genotypes on dependence of SU on FEUA; each SU and commensurate FEUA measured at each of 5 time points plotted for each individual with the Q141 allele (**d**) or 141K allele (**e**) with linear regression fits ($p < 0.0001$ for all fits) extended until FEUA = 0. Plot of the percentage of SU variability due to renal ($r^2$) or extra-renal ($1−r^2$) urate handling across increasing urate loads for Q141 participants (**f**) or 141K participants (**g**). **h** Effect of ABCG2 genotypes on dependence of SU on UUE at baseline (time 0) plotted with linear regression fits (Q141 allele, $p = 0.86$; 141K allele, $p = 0.02$) extended until UUE = 0. Source data are provided as a Source data file.

$p < 0.0001$ for all fits). Interestingly, a comparison of the dependence of SU on FEUA, as measured by the slope of the linear regression fits, showed no differences between the Q141 and 141K participants at baseline, but did show a significant difference after 180 min (ANCOVA, $p = 0.0005$, Supplementary Fig. 1b). This suggested that as filtered urate load dynamically increased, a significant difference emerged between the Q141 and 141K participants in the relationship between FEUA and SU.

We then examined the linear regression fits at each time point for each genotype (Fig. 1d, e). In the Q141 individuals, there was a significant change in the slope of the dependence over time (ANCOVA, $p = 0.0025$). Extending the linear regression of each time point in order to model the hypothetical SU with zero FEUA (no renal excretion of urate) revealed the SU would remain ≤8 mg/dL in Q141 individuals at the SU loads achieved (Fig. 1d). In the 141K individuals, the dependence of SU on FEUA remained unaltered through time, with only a significant change in the elevation (ANCOVA, $p < 0.0001$). These data hint at a loss of excretion capacity. This hypothesis is supported by modeling where FEUA is zero, revealing an increasing SU over time such that at 180 min the hypothetical SU with zero FEUA was ~11 mg/dL (Fig. 1e). To understand the contribution of renal and extra-renal function to the variability observed in SU, we plotted the coefficient of determination ($r^2$, see "Methods" section) and non-determination ($1-r^2$) versus SU (Fig. 1f, g). In Q141 participants, the extra-renal contribution was greater than the renal contribution to the observed variability in SU, over time and increasing urate load. Surprisingly, in the 141K individuals, the contribution of renal function was greater than extra-renal contributions (Fig. 1g). A similar comparison of the relationship between SU and UUE at the beginning of urate loading showed that UUE was not correlated with SU in the Q141 individuals (Fig. 1h), but was correlated in those with the 141K variant (Pearson, $p = 0.02$). Together, these analyses demonstrate that in individuals with the 141K variant, urate handling and excretion via the kidney was more important in determining SU than in Q141 individuals. This demonstrates a significant role for extra-renal excretion in determining SU in our study population, and critically, a significantly altered role in 141K participants. Although the interventional trial data suggested a significant role for ABCG2 in extra-renal urate excretion, we did not observe an altered FEUA in the 141K individuals. It was unclear whether this was because our study was underpowered to resolve small FEUA changes, ABCG2 plays no significant role in renal urate handling in humans, or that the Q141K variant has no discernable effect on ABCG2 function in human kidneys.

**ABCG2 in human and mouse kidneys**. The role of ABCG2 in human kidney function is controversial. Therefore, we conducted our own investigation into the presence of ABCG2 protein in the human kidney and found ABCG2 present in the cortex (Fig. 2a, b) and medulla (Fig. 2g–j; Supplementary Fig. 3a), colocalized with the proximal tubule marker *Lotus tetragonolobus* lectin (LTL). Surprisingly, the number of tubules with ABCG2 signal appeared greater in the sections taken from the outer stripe of the medulla (Fig. 2g–j), suggesting a prominent role in the straight (S3) portion of the proximal tubule. The immunofluorescence signal corresponded well with western blot analysis of the same tissue (Fig. 2e). ABCG2 localization matched localization of URAT1 (*SLC22A12*) by western blot studies and localization in LTL-positive proximal tubules of the renal cortex (Fig. 2c, d, f). The localization of cortical intratubular and intracellular URAT1 does not directly overlap with the LTL marker, but is more apical. In contrast, ABCG2 substantially colocalized with LTL, an observation consistent with a slight difference in the localization of ABCG2 and URAT1 in the cells of the cortical proximal tubule. ABCG2-positive cytosolic puncta were also observed in a number of medullary tubules (Fig. 2g–j). Primary human cortical renal epithelial cells cultured on transwells (Supplementary Fig. 3b) showed significant amounts of ABCG2 protein within the apical compartment/brush border (above the tight junctions, as demarcated by zonula occludens 1 [ZO-1], Fig. 2k). However, there was also a significant portion of the ABCG2 signal localizing

to puncta in the cytosol. This contrasted strongly with what was observed with URAT1, which was found exclusively above the nucleus in the apical/brush border compartment (Fig. 2l). This complexity of ABCG2 abundance and localization in human renal epithelia suggests an important role for the regulation of trafficking of renal ABCG2 in urate handling.

For comparison, in the mouse kidney, ABCG2 is highly abundant (Fig. 2m, Supplementary Fig. 3c) and localizes to the S2–S3 portion of proximal tubule (identified as LTL-positive) similar to what was observed in the human kidney tissue. Intratubular and intracellular localization showed that ABCG2 is abundant on the apical brush border surface, apical to the LTL staining (Fig. 2n, o), similar to URAT1 (Fig. 2p–r), but lacked the intracellular localization observed in the human renal tissue. Intriguingly, neither ABCG2 nor URAT1 strongly localize in the early proximal tubule (S1), the proposed site of the majority of urate reabsorption (Fig. 2s).

**Q140K+/+ hyperuricemic mouse model**. To better address the contradictory evidence in humans concerning the pathogenic mechanisms of the Q141K ABCG2 variant, we created an orthologous knock-in of the Q141K human variant in the C57BL6J mouse background using CRISPR-Cas9 genome editing (Fig. 3a, methods). The mouse and human ABCG2 proteins are very similar (81% homology), with the human glutamine at position 141 homologous to mouse ABCG2 Q140. The resulting homozygous animals (Q140K+/+) bred successfully, had no outward phenotypic alterations, and no effect on survival (80-week survival: WT 80%, $n = 10$; Q141K+/+ 94%, $n = 18$). The male Q140K+/+ animals displayed a significant increase (+89.4%; $p = 0.0003$, Student's $t$-test) in SU (Fig. 3b). This robust increase in SU was observed in mice on normal chow and with no disruption of uricase function, suggesting that physical transport of urate plays an extremely important role in determining SU. Interestingly, female mice of comparable age showed no alterations in SU (Fig. 3c and Supplementary Fig. 4), mirroring the observed sex differences in SU association with the human Q141K variant (SNP rs2231142)[14,24].

For evaluation of kidney function, we used metabolic cages to collect urine over a 24 h period with terminal serum, and measured electrolytes, creatinine, and urate (Table 1). Among other differences, we observed an increase in the fractional excretion of sodium (FeNa+) and a decrease in glomerular filtration rate (GFR) in the Q140K+/+ animals, observations potentially related through tubuloglomerular feedback. As with SU, investigation of the female Q140K+/+ mice chemistries demonstrated few of the differences observed in the male mice (Supplementary Table 2), providing further support to sexual dimorphism in the pathogenicity of the Q141K variant. The calculated fractional excretion of urate (FEUA) in the WT male mice was $3.88 \pm 0.67\%$ (± standard error of the mean [SEM]; $n = 9$, Fig. 3c) relatively similar to humans (5.68%) at baseline. A comparison of the dependence of SU on kidney urate handling (FEUA) in humans and WT male mice (Fig. 3d) found a remarkable similarity in their dependence of SU on FEUA; only the elevation was altered (ANCOVA, $p < 0.0001$), a difference explained by the substantial difference in urate load. FEUA in the male Q140K+/+ mice was significantly decreased (Fig. 3c; 47%, $n = 12$, $p = 0.01$; no change in female mice, Supplementary Fig. 4, $n = 7$, $p = 0.6263$, Student's $t$-test), supporting the role of ABCG2 in the secretion of urate. The reduced FEUA coupled with an increased urate load resulted in our inability to observe a significant increase in UUE (Fig. 3e), though our UUE measurements had a large variability, suggesting that the alteration in renal excretion was not responsible for the

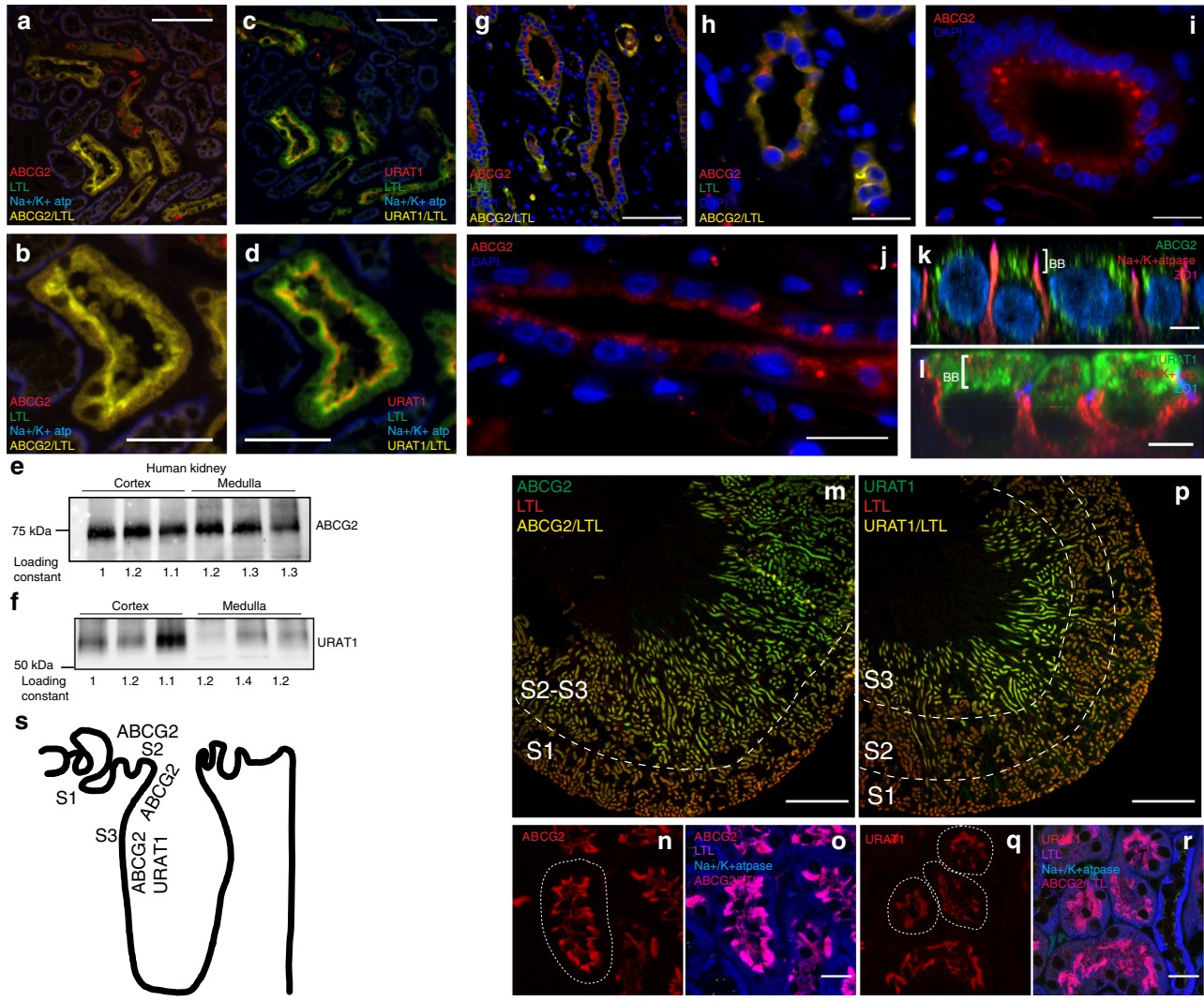

**Fig. 2 Renal ABCG2 localization in the human and mouse proximal tubule. a, b** Representative immunofluorescence micrographs of fixed cortical sections of human kidney samples (Q141) stained for ABCG2 (red), the proximal tubule marker LTL (green), the basolateral marker Na+/K+ ATPase (blue), and colocalization of ABCG2 and LTL (yellow) at 200× (**a**; scale bar 100 μM) and 400× (**b**; scale bar 50 μM). **c, d** Serial sections (from **a, b**) stained for URAT1 (red), LTL (green), Na+/K+ ATPase (blue), and colocalization of URAT1 and LTL (yellow) at 200× (**c**) and 400× (**d**) (**a–d** representative of three independently repeated experiments from $n = 3$ kidneys). Western blots from cortical samples ($n = 3$ kidneys) and medullary samples ($n = 3$ kidneys) for ABCG2 (**e**) and URAT1 (**f**). Representative immunofluorescence micrographs of fixed medullary sections of human kidney samples stained for ABCG2 (red), LTL (green), DAPI (blue), and colocalization of ABCG2 and LTL (yellow) at 200× (**g**; scale bar 100 μM) and 400× (**h**; scale bar 25 μM), or 400x without LTL (**i, j**; scale bar 25 μM) (**g–i**, representative of three independently repeated experiments from $n = 3$ kidneys). Z projections of cultured human renal epithelial cells stained for **k** ABCG2 (green), Na+/K+ ATPase (red), and ZO-1 (magenta), or **l** URAT1 (green), Na+/K+ ATPase (red), and ZO-1 (blue)(scale bar 5 μM). Representative (similar staining was repeated on tissue from $n = 3$ mice) micrographs (40×; scale bar 500 μM) from fixed whole mouse kidney slices stained with ABCG2 (**m**) or URAT1 (**p**) in green, LTL (red), and colocalization (yellow). Higher magnification (400×; scale bar 20 μM) micrographs on fixed sections from mouse kidney stained for ABCG2 (**n, o**) or URAT1 (**q, r**) in red, and in **o, r** LTL (magenta), Na+/K+ ATPase (blue), and colocalization of ABCG2/URAT1 and LTL (pink). **s** Diagram of transporter localization. Source data are provided as a Source data file.

hyperuricemia observed in the Q140K+/+ animals. A comparison of differences in SU dependence on FEUA in WT and Q140K+/+ male mice found, as observed in the human population, that SU is significantly dependent on FEUA (linear regression, WT $p = 0.020$; Q140K+/+ $p = 0.033$). Fit comparisons revealed significantly altered slopes (ANCOVA, $p = 0.0083$) (Fig. 3f), and comparison of the coefficients of determination ($r^2$) showed that in WT animals 51% of SU variation is explained by renal excretion (49% by extra-renal excretion), but the renal contribution is increased to 72% in Q140K+/+ animals (28% by extra-renal) (Fig. 3g).

A relationship between hyperuricemia and metabolic alterations has been previously hypothesized[1]. We, therefore,

investigated potential changes in key markers of metabolic health in the Q140K+/+ mice. We found significant increases in serum glucose ($p = 0.003$, Student's $t$-test), insulin ($p = 0.018$, Student's $t$-test), and insulin-like growth factor 1 (IGF1) ($P = 0.041$, Student's $t$-test), but not in body mass ($p = 0.15$, Student's $t$-test) (Fig. 4a). Next, we observed sporadic fatty liver phenotypes in Q140K+/+ males, but not WT animals (Fig. 4b and Supplementary Fig. 5) and found in the same males altered gene expression in markers of fatty liver disease (Pnpla2, $p = 0.012$; Pnpla3, $p = 0.050$; and G6pd, $p = 0.032$, Student's $t$-tests; Fig. 4c and Supplementary Fig. 5). The female Q140K+/+ mice were not hyperuricemic, and had no alterations in glucose, insulin, or IGF1 (Supplementary Table 2). The observation of significant

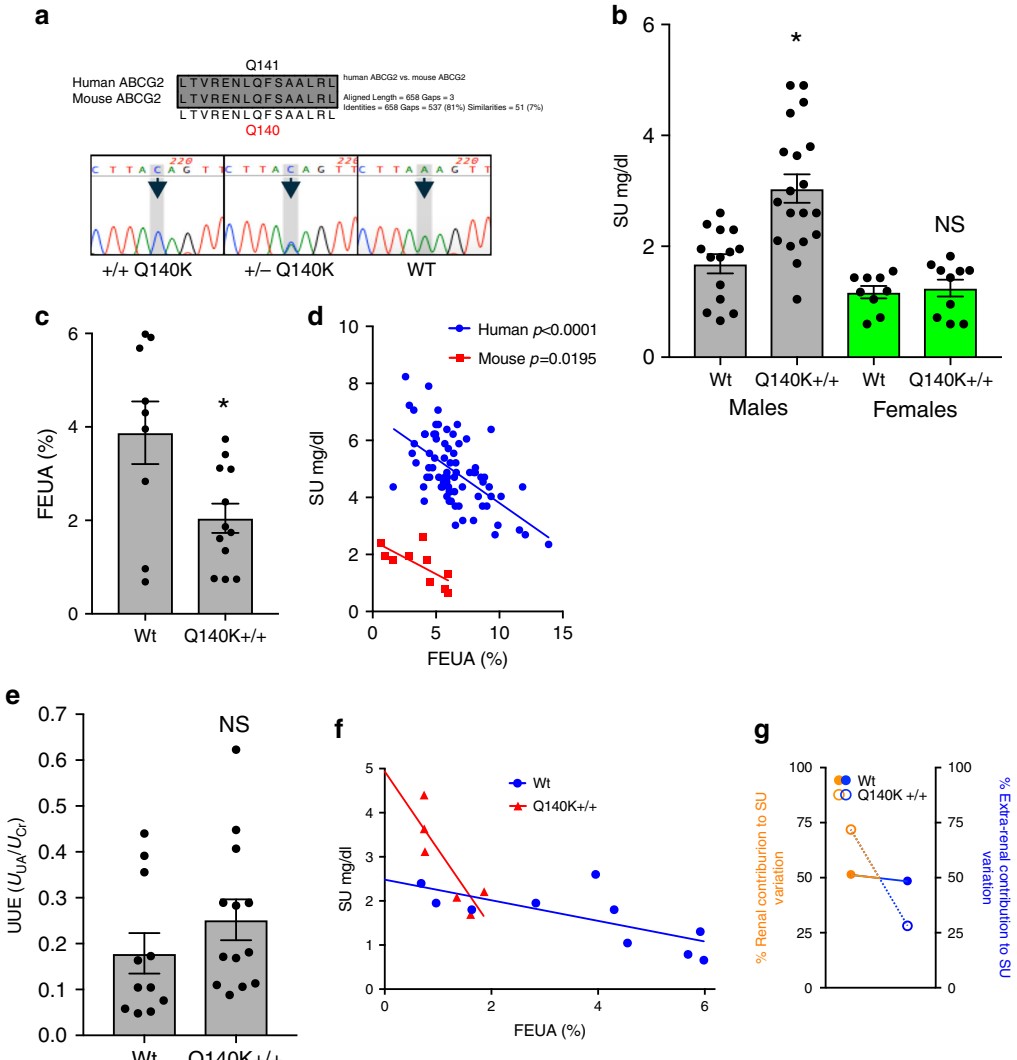

**Fig. 3 Male Q140K+/+ mice are hyperuricemic and demonstrate a renal urate handling phenotype similar to humans with 141K allele. a** Alignment of human and mouse ABCG2 amino acid sequence in region near Q141 residue, and the color-gram of sequencing results of animals heterozygotic for the alterations to the endogenous *Abcg2* loci or homozygotic. **b** Serum urate levels for male and female, WT and Q140K+/+ animals (WT males $n = 14$; Q140K+/+ males $n = 19$; $p = 0.0003$; WT females n = 9; Q140K+/+ $n = 10$; $p = 0.7113$). **c** Fractional excretion of urate (FEUA) measurements from male WT ($n = 9$) and male Q140K+/+ ($n = 12$) ($p = 0.01$). **d** Comparison of Q141 human participants ($n = 79$) to WT mice ($n = 10$) and their respective relationship between FEUA and SU, fit with linear regression that are not different in slope ($p = 0.6390$), but are different in elevation ($p < 0.0001$). **e** Urinary urate excretion as a function of urinary creatinine was not significantly different between WT ($n = 11$) and Q140K+/+ ($n = 13$) mice ($p = 0.259$). **f** Dependence of SU on FEUA plotted for each WT ($n = 10$) or Q140K+/+ ($n = 6$) is significant for both genotypes (WT $p = 0.0195$; Q140K+/+ $p = 0.0330$), but the slope of the linear regression fits are significantly different ($p = 0.0083$). **g** Plot of the percentage of SU variability due to renal ($r^2$) or extra-renal ($1-r^2$) urate handling in WT or Q140K+/+ mice. Statistical analysis: **b**, **c**, **e** two-tailed Student's $t$-test, ± SEM; **d**, **f** Linear regression with slope and elevation comparison using two-tailed ANCOVA. Source data are provided as a Source data file.

differences in liver phenotypes and markers of metabolic health in the Q140K+/+ animals suggest a direct connection between ABCG2 function, SU, and glucose metabolism.

**ABCG2 abundance and localization in Q140K+/+ mouse kidney.** The role of reduced ABCG2 protein abundance in explaining altered kidney urate handling in the male Q140K+/+ animals was assessed. We found no difference in ABCG2 mRNA levels between the two genotypes (Fig. 5a). Immunofluorescence investigation of whole kidney slices (Fig. 5b) revealed subtle, but significant differences in ABCG2 immunofluorescent signal of the mutant ABCG2 protein in both the S1 ($p = 0.02$, Student's $t$-test) and the S2 segments ($p = 0.0008$, Student's $t$-test) (Fig. 5c).

Higher magnification comparison of individual tubules showed no change in the brush border/apical localization of mutant ABCG2 in the proximal tubule (Fig. 5d), but confirmed the subtle yet significant decrease in ABCG2 immunofluorescent signal ($p = 0.0007$, Student's $t$-test, Fig. 5e). Western blot of the male whole kidney lysate confirmed the reduced total abundance of ABCG2 Q140K+/+ protein ($p < 0.0001$, Student's $t$-test, Fig. 5f), but with no statistical difference in the heterozygotic male animals (Supplementary Fig. 2), or females (Supplementary Fig. 4). Critically, we found that FEUA showed a significant correlation (Pearson, $p = 0.0033$) and dependence (linear regression, $p = 0.0066$) on renal ABCG2 abundance (both WT and Q141K) confirming a key role for even the mutant protein on renal handling of urate (Fig. 5g). Finally, we investigated the

**Table 1 Blood and urine chemistries of male WT and Q140K+/+ mice.**

| | Blood chemistry | | | Urine chemistry | |
|---|---|---|---|---|---|
| | WT (n = 8ᵃ) | Q141K+/+ (n = 7ᵃ) | | WT (n = 6) | Q141K+/+ (n = 6ᵇ) |
| Na+ [mmol/l] | 144.125 ± 1.48 | 144.71 ± 0.95 | Na+ [mmol/l] | 39.60 ± 2.92 | **49.02 ± 3.85 [p = 0.048]** |
| K+ [mmol/l] | 4.60 ± 0.18 | **5.17 ± 0.19 [p = 0.027]** | K+ [mmol/l] | 90.67 ± 5.96 | 99.70 ± 4.81 |
| Cl− [mmol/l] | 114.88 ± 1.54 | 111.71 ± 0.91 | Cl- [mmol/l] | 90.77 ± 9.23 | 107.75 ± 8.21 |
| BUN [mg/dl] | 26.29 ± 0.78 | 24.57 ± 1.97 | FENa+ [%] | 0.47 ± 0.06 | **0.73 ± 0.1 [p = 0.024]** |
| Hct [%PCV] | 35.50 ± 0.88 | 38.14 ± 1.32 | FEK+ [%] | 33.5 ± 5.7 | 34.4 ± 3.2 |
| pH | 7.10 ± 0.03 | 7.18 ± 0.03 | FECl− [%] | 1.4 ± 0.25 | 1.8 ± 0.16 |
| PCO₂ [mmHg] | 62.08 ± 4.45 | 55.61 ± 2.63 | AnGap [mmol/l] | 39.50 ± 4.89 | 40.97 ± 9.70 |
| HCO₃ [mmol/l] | 19.44 ± 0.78 | 20.21 ± 0.68 | UV [ml/24 h] | 4.49 ± 0.69 | 3.94 ± 0.32 |
| Beecf [mmol/l] | −11.50 ± 0.58 | **−8.83 ± 1.08 [p = 0.038]** | Osmolarity [mmol/kg] | 808.33 ± 61.03 | **973.67 ± 65.08 [p = 0.047]** |
| AnGap [mmol/l] | 17.33 ± 0.49 | **21.00 ± 1.67 [p = 0.02]** | | WT (n = 9) | Q141K+/+ (n = 11) |
| Hb [g/dl] | 12.09 ± 0.30 | 12.97 ± 0.37 | GFR | 207.2 ± 30.0 | **150.9 ± 7.8 [p = 0.031]** |

Blood, serum, and urine chemistries from WT and Q140K+/+ male animals. Blood chemistries acquired using the iSTAT device (see "Methods" section).
Bold values are significant; statistical analysis: Student's *t*-tests; ±SEM.
*BUN* blood urea nitrogen, *Hct* hematocrit, *Beecf* base excess, *AnGap* anion gap, *Hb* hemoglobin, *FE* fractional excretion, *UV* urinary volume (rate), *GFR* estimated glomerular filtration rate.
ᵃGroup variability; WT: K+ n = 6; BUN n = 7; Beecf/AnGap n = 6; Q140K+/+: Beecf n = 6; AnGap n = 5.
ᵇGroup variability; Q140K+/+: FENa+ n = 4; FEK+/ FECl− n = 5.

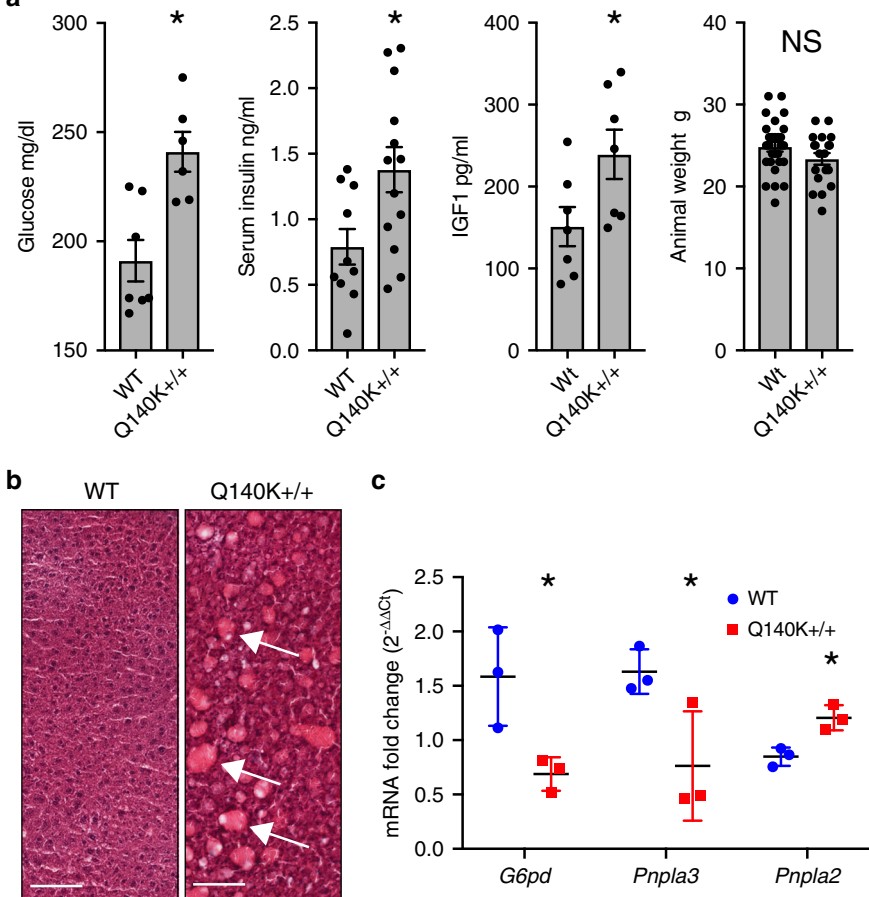

**Fig. 4 Male Q140K+/+ mice are hyperglycemic and develop fatty livers. a** Serum levels of glucose (WT *n* = 7; Q140K+/+ *n* = 6; *p* = 0.003), insulin (WT *n* = 10; Q140K+/+ *n* = 13; *p* = 0.018), insulin-like growth factor 1 (IGF1) (WT *n* = 7; Q140K+/+ *n* = 7; *p* = 0.041), and male animal mass (WT *n* = 28; Q140K+/+ *n* = 19; *p* = 0.15) (all ± SEM). **b** H and E stained representative liver samples from WT and Q140K+/+ male mice; fatty deposits indicated with white arrows (representative of six independent experiments; *n* = 6 WT and *n* = 6 Q140K+/+ livers; scale bar 200 μM). **c** Quantitative real-time PCR analysis of total liver mRNA from WT and fatty Q140K+/+ livers (*n* = 3 for each) show significant differences in mRNA from genes associated with fatty liver disease; *G6pd* (*p* = 0.032), *Pnpla3* (*p* = 0.050), and *Pnpla2* (*p* = 0.012) (±SEM). Statistical analysis: **a, c** two-tailed Student's *t*-test. Source data are provided as a Source data file.

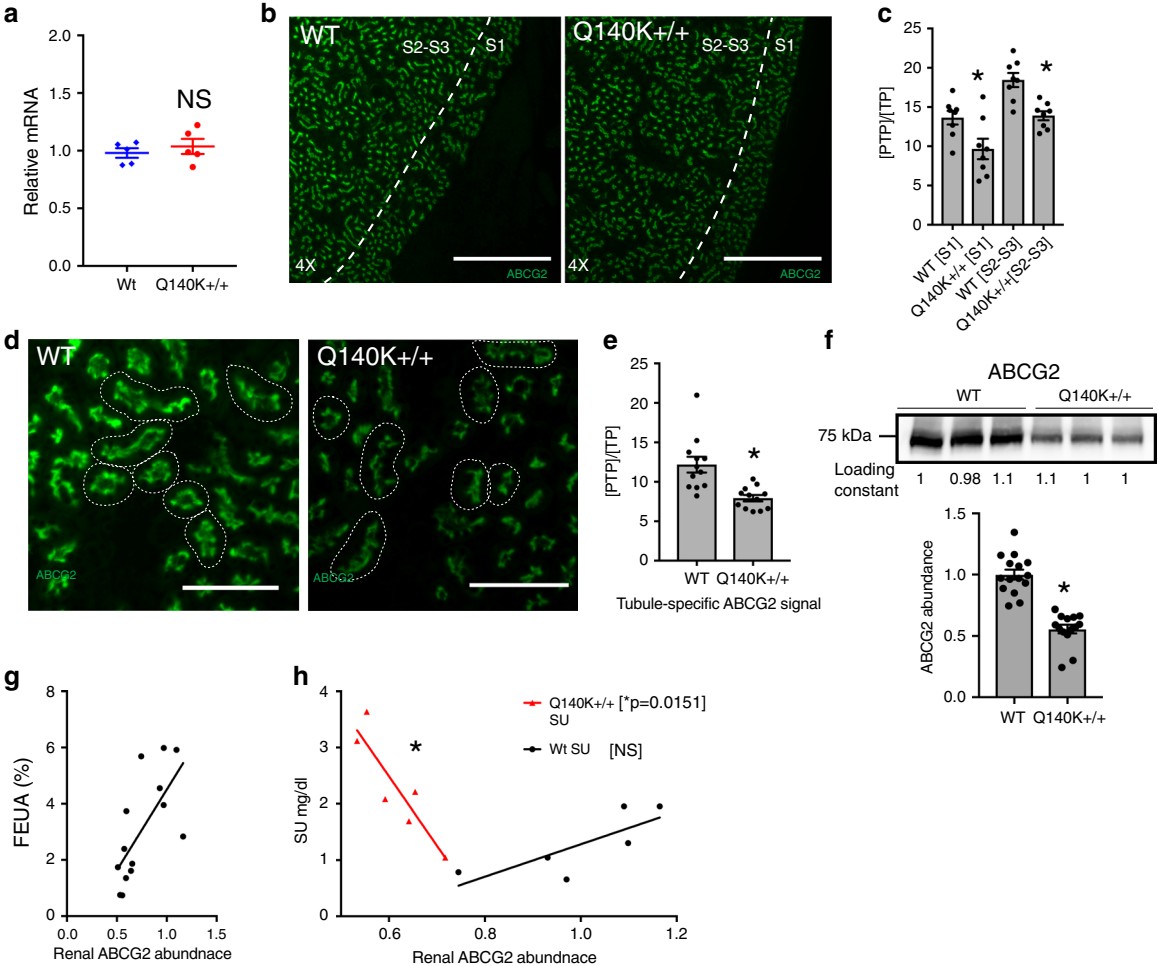

**Fig. 5 Subtle alterations of ABCG2 abundance and localization in Q140K+/+ mouse kidney. a** Quantitative real-time PCR analysis of total kidney mRNA shows no significant difference in the mRNA of ABCG2 in WT as compared to Q140K+/+ ($n = 5$ for both, $p = 0.4848$; ± SEM). **b** Representative (similar staining was repeated on tissue from $n = 3$ mice of each genotype) micrographs (scale bar 500 μM) from fixed whole mouse kidney slices stained with ABCG2 (green). **c** Quantification of immunofluorescence ABCG2 signal in either S1 or the S2–S3 segments of WT and Q140K+/+ show significant differences (metric = positive thresholded pixels / total pixels [PTP/TP], see "Methods" section), S1: $p = 0.02$; $n = 8$ analysis areas from kidney sections from 3 WT animals and 3 Q140K+/+ animals; S2–S3: $p = 0.0008$; $n = 8$ analysis areas from kidney sections from 3 WT animals and 3 Q140K+/+ ; ±SEM. **d** Representative micrographs (scale bar 100 μM) at higher magnification to visualize individual tubules, stained with ABCG2 (green). **e** Quantification of immunofluorescent ABCG2 signal in individual tubules from WT and Q140K+/+ mice show tubule specific significant differences in [PTP/TP] ($p = 0.0007$; n = 12 analysis areas from kidney sections of 3 animals of both genotypes; ±SEM. **f** Western blots of total kidney homogenate from WT ($n = 15$) and Q140K +/+ ($n = 14$) male mice showed a significant decrease in abundance of the Q140K+/+ protein ($p < 0.0001$; loading constant determined by total lane protein analysis [methods]; ±SEM). **g** FEUA showed a significant correlation (Pearson test statistic, $p \leq 0.0033$) and dependence (linear regression, $p = 0.0066$) on renal ABCG2 abundance (both WT and Q140K+/+). **h** Plotted dependence of SU on renal abundance of ABCG2 (WT $n = 6$; Q140K+/+ $n = 6$) with linear regression fits (WT no different than zero, $p = 0.07$; SU is significantly dependent on renal ABCG2 abundance in Q140K+/+ animals; $p = 0.0151$, $r^2 = 0.8064$). Statistical analysis: **a**, **c** two-tailed Student's $t$-test. Source data are provided as a Source data file.

relationship of renal ABCG2 abundance to the overall SU phenotype, and found renal ABCG2 abundance in the WT animals did not significantly correlate with SU (Fig. 5h), suggesting that small changes in ABCG2 abundance left sufficient remaining capacity to handle the low urate loads present with normal gastrointestinal urate excretion. Conversely, in the Q140K+/+ mice, SU showed a significant dependence on renal ABCG2 abundance (linear regression, $p = 0.015$, $r^2 = 0.81$; Fig. 5h), demonstrating that in the absence of significant gastrointestinal excretion and higher urate loads, the remaining ABCG2 capacity in the kidney is critical for setting the SU.

**Alterations in intestinal urate secretion in Q140K+/+ mice.** Our data from humans with the 141K allele and Q140K+/+

mice, as well as previous work documenting the role for ABCG2 in gastrointestinal excretion of urate[15,16,31], led our focus to investigating gastrointestinal-mediated urate excretion defects in the Q140K+/+ mouse model. First, we found the highest ABCG2 protein abundance in the jejunum and ileum of the small intestine; a finding similar to previous mRNA expression studies[32,33] (Fig. 6a). Immunofluorescence investigation of mouse intestinal sections from the jejunum and ileum localized ABCG2 protein to the apical brush border of the villi cells (Fig. 6b). As in the kidney, ABCG2 apical localization in the villi cells supports the role of ABCG2 secretion of urate into the lumen of the intestines. ABCG2 also localized to the apical surface of crypt cells although with a less obvious physiological consequence (Supplementary Fig. 6a). We found that SLC2A9 strongly colocalizes with the alpha subunit of Na+/K+ ATPase (Fig. 6c) revealing a prominent

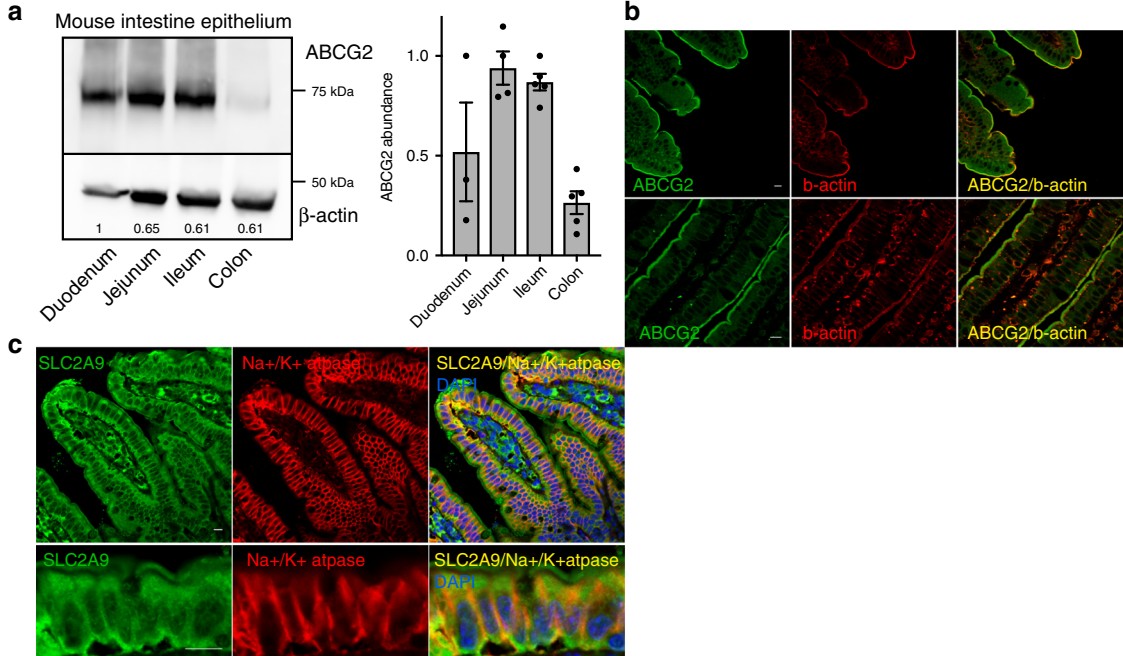

**Fig. 6 Localization of ABCG2 in mouse intestine epithelium. a** Representative western blot of homogenized dissected intestinal segments from WT mice (duodenum $n = 3$; jejunum $n = 4$; ileum and colon $n = 5$) showed greatest ABCG2 abundance in the small intestine jejunum and ileum segments; beta-actin used as loading control; summary data of ABCG2 abundance. **b** Representative immunofluorescence micrographs of fixed sections of mouse small intestines stained for ABCG2 (green), villi brush border marker beta-actin (red), and colocalization of ABCG2 and beta-actin (yellow) at 200× (all scale bars 10 µM) and 400x or **c** stained for SLC2A9 (green), Na+/K+ ATPase (red), DAPI (blue), and colocalization of SLC2A9 and Na+/K+ ATPase (yellow) 400×, with Z projections (all scale bars 10 µM) (**b**, **c** representative images of sections from $n = 4$ mice). Source data are provided as a Source data file.

basolateral localization consistent with the hypothesized role of SLC2A9 in the intestinal urate secretory pathway[17].

A comparison of localization and abundance of ABCG2 in the WT and Q140K+/+ animals revealed significant differences. First, total ABCG2 protein abundance, but not mRNA levels, in the jejunum was significantly and dramatically reduced in the Q140K+/+ animals ($p = 0.0046$, Student's $t$-test, Fig. 7a, b, Supplementary Fig. 6b). The reduction in protein abundance is far greater than that observed in the kidney of the Q140K+/+ animals (78% vs 44%). The severity of the Q140K ABCG2 protein abundance loss was also observed in a significantly reduced signal in the immunofluorescence of the jejunum of Q140K+/+ mice compared to WT controls ($p = 0.018$, Student's $t$-test, Fig. 7c).

Using an ex vivo intestinal loop assay (Fig. 7d, see "Methods" section), we assessed the effects of Q140K on ABCG2-mediated urate flux. In WT male loops, a net secretory urate flux was observed that was concentration dependent (Fig. 7e), and highly sensitive to a luminal ABCG2 inhibitor topiroxostat[34] (TPX, $p = 0.0019$, Student's $t$-test, Supplementary Fig. 6c). Comparing WT and Q140K+/+ animals revealed that the Q140K+/+ mice had significantly decreased basolateral to luminal urate flux ($p < 0.0001$, Student's $t$-test, Fig. 7f), a large enough decrease to reverse the net secretion of urate to net absorption in the Q140K+/+ intestinal loops (Fig. 7h). Subtraction of the calculated flux sensitive to ABCG2 inhibitor, TPX, revealed about half of WT urate flux is mediated by a TPX-sensitive pathway, but there is close to zero TPX-sensitive flux in the Q140K+/+ loops (Fig. 7g). This result is consistent with an almost complete loss of ABCG2-mediated urate secretion in the Q140K+/+ mice. There is no change in urate flux in Q140K+/+ colon derived loops as compared to WT (Fig. 7f), consistent with little ABCG2-mediated secretion in the large intestines. The intestinal experiments in the male Q140K+/+ mouse model support a near complete loss of ABCG2-mediated intestinal secretion of urate in contrast to a largely preserved renal function in the mutant Q140K+/+ animals. Therefore, the loss of intestinal urate secretion appears to drive the hyperuricemia in the Q140K+/+ ABCG2 mice and potentially in the 141K variant carriers in our clinical cohort.

## Discussion

Three transporter genes *SLC2A9*, *SLC22A12*, and *ABCG2* have the largest contribution to risk for hyperuricemia and gout, but where the protein products for these genes, SLC2A9/Glut9, URAT1, and ABCG2, respectively, localize, where they are important for urate excretion, and how they contribute to urate homeostasis remains largely undefined. Here, we have focused on the role of *ABCG2*, the highest risk gout gene[14,24], in urate excretion. Previously, Matsuo et al.[31] correlated levels of ABCG2 dysfunction with alterations in both intestinal excretion (renal overload hyperuricemia) and renal under-excretion, demonstrating that even moderate ABCG2 dysfunction may contribute to renal under-excretion. However, in human studies of individuals with at least one copy of the common *ABCG2* variant 141K, but not enriched for gout, there has been conflicting conclusions on the effect of renal urate excretion (FEUA)[24–26]. Even in the studies reporting a significant difference in FEUA associated with the 141K allele[24], the difference is very small (0.217 mg/dl), and thus physiological relevance is uncertain. So, what is the quantitative role for ABCG2-mediated urate excretion in the kidney and the gastrointestinal tract?

Here, we approached this problem from two perspectives. First, we used a human interventional trial to assess the contribution of variant genotypes in individuals without gout on renal excretion following a purine challenge. We observed a significantly higher SU in individuals with the 141K allele at baseline and at each time point after the oral inosine load, observations that support the role for ABCG2 in secretion of urate. However, we did not observe any differences in the FEUA or UUE for individuals with

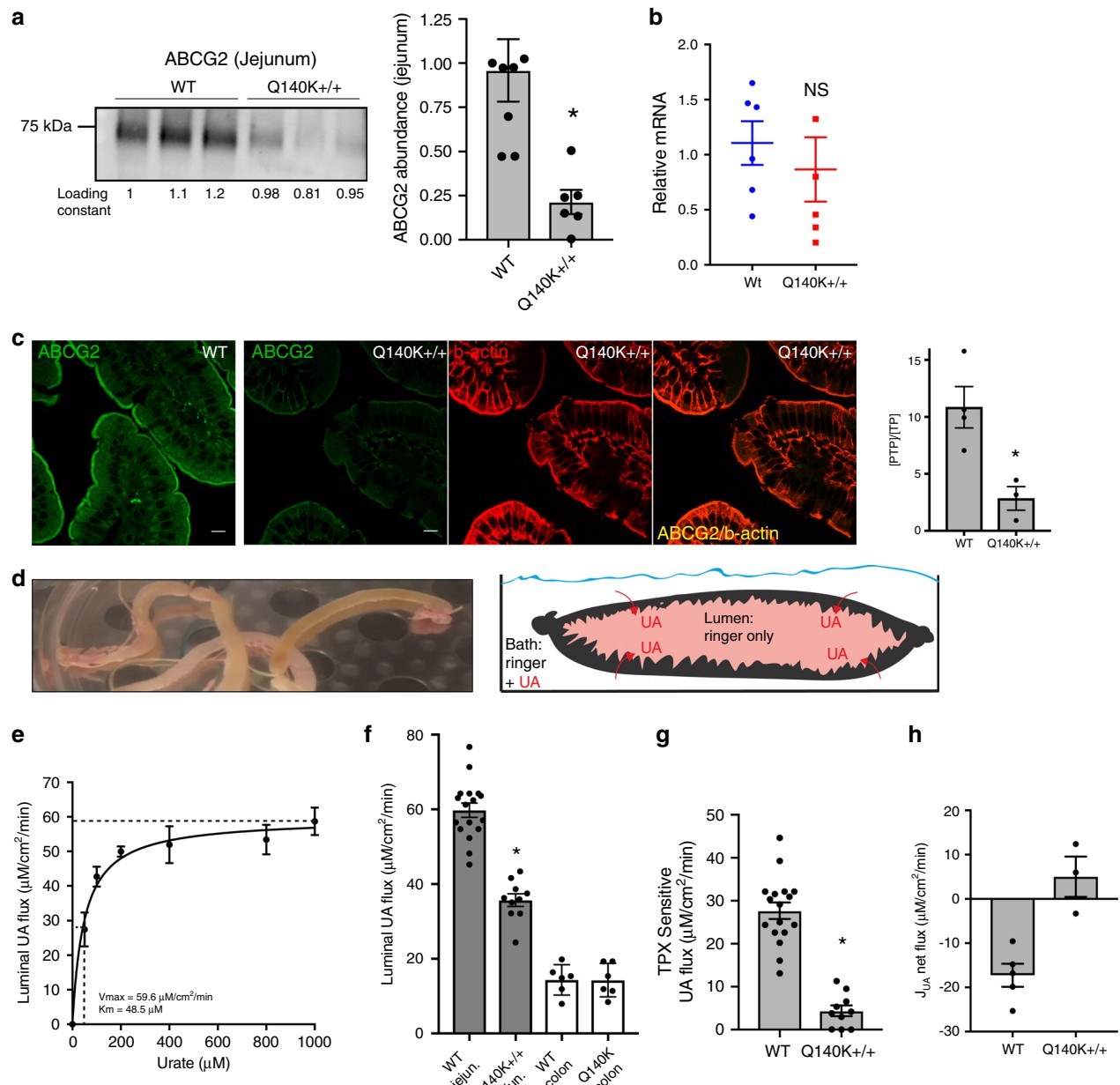

**Fig. 7 Severe loss of urate secretion in the intestines of Q140K+/+ mice. a** Western blots of jejunum homogenate from WT (*n* = 8) and Q140K+/+ (*n* = 6) mice showed a significant 78% decrease in abundance of the Q140K+/+ protein (*p* = 0.0046; loading constant determined by total lane protein analysis; ±SEM). **b** Quantitative real-time PCR analysis of small intestine mRNA shows no significant difference in the mRNA of ABCG2 in WT as compared to Q140K +/+ (*n* = 5 for both, *p* = 0.5132; ±SEM). **c** Representative immunofluorescence micrographs of fixed sections of mouse small intestines (from *n* = 3 WT and *n* = 3 Q140K+/+ mice) stained for ABCG2 (green), villi brush border marker beta-actin (red), and colocalization of ABCG2 and beta-actin (yellow) at 200×(all scale bars 10 µM); quantification of immunofluorescent ABCG2 signal in either WT and Q140K+/+ jejunum show significant differences in [PTP/TP] (*p* = 0.018; *n* = 4 analysis areas from jejunum sections from 4 WT animals and *n* = 3 analysis areas from 3 Q140K+/+ animals; ±SEM). **d** Image of intestinal loops prepared from mouse small intestines; and graphic demonstrating the functional lumen urate accumulation assay. **e** Plot of urate concentration versus luminal UA flux fit successfully with a Michaelis–Menten curve, Vmax = 59.58 ± 2.74 µM/cm²/min; Km = 48.45 ± 11.44µM, *n* = 5 animals for each of seven urate concentrations; ±SEM. **f** Mean luminal flux in WT (*n* = 17) and Q140K+/+ (*n* = 10) jejunum loops were significantly different (*p* < 0.0001); and in colonic loops (*n* = 6 for both; *p* = 0.974). **g** Modeled ABCG2 mediated flux (mean ABCG2 inhibitor TPX sensitive flux subtracted, see "Methods" section) demonstrate almost complete loss in Q140K+/+ animals (reduced 84.2%; WT *n* = 17 and Q140K+/+ *n* = 10, *p* < 0.0001; ±SEM). **h** Calculated jejunum urate net transport (see "Methods" section) shifts from net luminal secretion to net absorption in the Q140K+/+ loops (WT *n* = 5; Q140K+/+ *n* = 3; *p* = 0.0036; ±SEM). Statistical analysis: **a–c**, **f–h**, two-tailed Student's *t*-test. Source data are provided as a Source data file.

the 141K allele in comparison to the Q141 individuals. These FEUA data are similar to those reported previously[26], but they also were surprising given the increase in SU in these individuals. More significantly, we observed a critical genotype difference in the role of FEUA in determining the SU, with individuals with the

141K allele losing the extra-renal contribution to excretion of urate. These data are consistent with the previous observations of loss of gastrointestinal-mediated urate secretion (renal overload type) in individuals with ABCG2 dysfunction[16,31]. Surprisingly, we did not see a genotype difference in UUE, a previously

described component of the renal overload hypothesis[16,31]. This may be explained by significant differences in the characteristics and genetic background of the study participants. We recruited young and healthy volunteers of both sexes, in contrast to the hyperuricemic and gouty participants of previous association studies[16,31]. Males and females were not segregated in our analysis, but a secondary analysis showed differences in UUE for the male and female non-carriers (Supplementary Fig. 7). However, analysis with a SNP*sex interaction term or female only analysis did not reveal any divergence from our segregated analysis (Supplementary Table 3 and Supplementary Fig. 7). Finally, lack of UUE differences in carriers of the Q141K risk allele may also be explained by genetic differences in the respective study populations, East Asian versus European, that have previously demonstrated differences for the association of the rs2231142 variant (Q141K) with poor allopurinol response[35,36]. An analysis of our data stratified by ancestry did not reveal substantial population differences between Polynesian and European populations.

The interventional trial also revealed more about how increased UA filtered loads are handled by the human kidney (Fig. 8). On the short time scale of the inosine load, we found that increased UA load led to incremental increases in FEUA and UUE, reflecting a passive spilling of UA into the urine with no energetic alterations to UA handling, and reabsorption capacity maximum at baseline. This raises the interesting possibility that under high urate filtered loads, the kidney risks high tubular urate concentrations and UA crystal deposition to protect a specific rate of UA reabsorption (FEUA < 10%), regardless of the SU level. The diminishment of renal contribution to SU in the Q141 participants as the filtered load increased revealed it is the extra-renal contribution that increases commensurate with the SU load to prevent significant alterations in SU (Fig. 8). This observation challenges the concept that the kidney is the primary regulator of baseline SU levels in humans.

Our second approach to understanding the pathology of the Q141K variant was to use a knock-in mouse model of the orthologous Q140K variant. Multiple attributes described here suggest mice present a reasonable model for understanding ABCG2-mediated urate handling in humans. The dependence of SU on FEUA is very similar, as is the prominent localization of ABCG2 protein in the S2–S3 segments of the proximal tubule. There are also significant similarities in the pathological physiology observed between the 141K individuals and the Q140K+/+ mice, but with important exceptions. The Q140K+/+ mice showed altered FEUA, whereas the humans with the 141K allele did not, a difference possibly explained by an allelic dose-response; our mice have two copies and the human participants have only one copy of the 141K allele (with the exception of a single TT individual). Importantly, the urinary excretion of urate is not significantly altered in the Q140K+/+ mice, but the role of the kidney in determining SU was, much like what we observed in the human interventional trial. An additional observation from the human interventional trial was that 141K individuals with high urate loads had an altered relationship between SU and FEUA—they lost the extra-renal contribution to the variation of SU and thus kidney handling of urate became the key determinant in setting absolute circulating urate levels. We observed the same phenomenon in the Q140K+/+ mice, allowing us to carefully dissect the causal mechanisms. We found these animals displayed an almost complete loss of ABCG2 protein and ABCG2-mediated intestinal excretion of urate, but with more subtle changes in renal protein abundance and function. Why does the renal urate excretion become more important in setting SU levels in the Q140K+/+ animals or 141K humans? Increased Q141K ABCG2 (Fig. 5h) abundance reduced the SU through

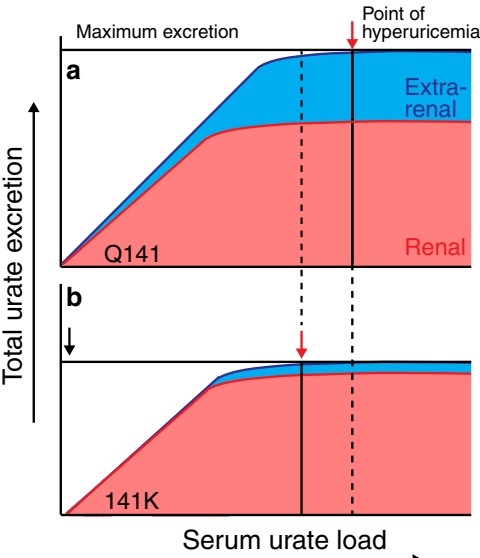

**Fig. 8 The Q141K variant affects extra-renal/intestinal urate excretion.** **a** Model of relationship between urate load and total urate excretion in Q141 individuals (and WT mice). As the urate load increases, total excretion increases equally, maintaining a serum urate set point. The total excretion is driven by renal excretion until high urate loads occur, and maximum renal excretion is reached. If the urate load continues to increase, increasing extra-renal excretion is necessary for serum urate set point maintenance. When total excretion reaches maximum, any additional urate load is retained, resulting in hyperuricemia (red arrow). **b** In 141K individuals (and Q140K+/+ mice), kidney excretion is largely maintained and is sufficient to balance moderate urate loads. However, as urate loads increase, 141K individuals have diminished extra-renal excretion, thus their maximum total excretion is significantly less (black arrow) and therefore smaller urate loads result in urate retention and hyperuricemia (red arrow).

increased secretion, demonstrating that the Q140K protein is still functionally very important, and with the blanket protection of the gastrointestinal excretion lost, small changes in ABCG2 abundance made big differences in SU.

An intriguing finding is the phenotype discrepancy between the male and female Q140K+/+ mice. Female mice do not exhibit hyperuricemia nor changes in FEUA. In addition, they show no alteration in Q140K protein abundance in the kidney, raising the possibility that being female alters the way the variant ABCG2 protein is trafficked and expressed, and thus for females the Q140K variant is essentially fully functional. These data are consistent with findings from human populations that show the Q141K ABCG2 variant has a stronger effect on urate in males than females[24]. Further understanding of why female mice are protected from the loss of function with the Q140K variant may illuminate how both ABCG2 and urate more generally is regulated in mammals.

Finally, the discrepancies of observed pathology between tissues may help explain the controversy over the role of ABCG2 in humans. Our confirmation of ABCG2 in human tubules and the Q140K+/+ mouse data identifies ABCG2 as important in renal handling of urate and supports previous conclusions[31] that ABCG2 plays a role in renal urate excretion. Our findings also propose an interesting solution as to why the *ABCG2* Q141K variant specifically accumulated in many modern human populations. The Q141K variant results in increases in SU, a likely selective advantage[37], but protects the renal handling of urate, without the risk of stones. Is it possible that the intricate scheme of reabsorption and secretion of urate along the proximal tubule is an evolutionary adaptation to urate per se, rather than simply

for the excretion of nitrogenous waste? Work[38] investigating the mechanism of observed increased urinary urate excretion in people with diabetes taking inhibitors of the sodium glucose 2 transporter (SGLT2) found that there is a relationship between tubular glucose and urate reabsorption though URAT1. It is intriguing there may be some role for the recycling of urate in regulating sodium/glucose reabsorption. In human studies, renal clearance of urate was shown to be inversely correlated with insulin resistance and directly correlated with the excretion of sodium[39]. In the Q141K+/+ mice, we observed significant hyperglycemia, significant increases in serum insulin, and IGF1, as well as increased fractional excretion of sodium. These data and the literature support a relationship between urate, glucose, and sodium handling along the nephron. In addition, the male Q140K+/+ mice presented fatty liver phenotypes[40] and expression changes of genes, including *Pnpla2* and *Pnpla3*, coding for triglycerol lipases implicated in human fatty liver disease[40–42], broadening the potential role of urate to disrupt metabolic processes globally[43]. Both intriguing possibilities deserve further exploration.

In conclusion, the minor allele of common human hyperuricemia and gout risk variant Q141K significantly alters the extra-renal contribution of urate excretion and SU determination in both humans and mice. Furthermore, Q140K+/+ knock-in mice have hyperuricemia, driven by an almost complete loss of ABCG2-mediated urate excretion in the small intestine. In contrast, the ABCG2 function in kidney is largely preserved indicating an intriguing differential pathology of the 141K variant. These conclusions provide support for a role of ABCG2 in human renal urate excretion, provide a better understanding of the respective contributions of the kidney and intestines to serum urate homeostasis, and finally, stimulate new interesting questions about the relationship between glucose, sodium, and urate transport.

## Methods

**Human interventional study**. This study was designed to examine the role of *SLC2A9* and *ABCG2* on serum urate and FEUA in response to an oral inosine load. Inosine is a purine nucleoside that is metabolized in vivo from AMP as part of the purine salvage metabolic pathway. Inosine is degraded to hypoxanthine, which is, in turn, metabolized to xanthine, and then urate in the purine degradation pathway. Administration of inosine leads to increased serum urate concentrations[44].

**Participants**. One hundred healthy participants were recruited by public advertising. Indigenous New Zealanders (Māori) and Pacific people living in New Zealand have very high prevalence of hyperuricemia and gout[45], and for this reason the sampling framework in the study protocol specified recruitment of 50 participants of Māori or Pacific ethnicity (Polynesian), and 50 participants of New Zealand European ethnicity. Inclusion criteria were: the ability to provide written informed consent, and estimated glomerular filtration rate (eGFR) >60 mL/min/ 1.73 m². Exclusion criteria were: ethnicity other than Māori, Pacific, or New Zealand European, first degree relative of another study participant (consistent with our prior short term intervention studies in which the primary aim was analysis of specific genetic variants), history of gout, history of kidney stones, history of diabetes mellitus, diuretic use, or urine pH ≤ 5.0. All visits took place at a clinical research center. The study was approved by the New Zealand Health and Disability Ethics Committee, and each participant gave written informed consent. The study was prospectively registered by the Australian Clinical Trials Registry (ACTRN12615001302549).

**Primary and secondary endpoints**. The primary endpoint was the change from baseline in serum urate concentration over 3 h following an inosine challenge. The secondary endpoint was change in fractional excretion of urate up to 3 h following an inosine challenge (the fractional excretion of urate is the ratio between the renal clearance of urate and the renal clearance of creatinine, expressed as a percentage) or the calculation of urinary urate excretion (UUE).

**Protocol**. Participants attended the study visit within two weeks of the screening visit. At the study visit, a venous catheter was inserted for blood collection. Following an overnight fast, participants took three 500 mg inosine tablets (Source Naturals) orally over a 5 min period between 0800 and 0930. Blood was obtained

for urate and creatinine prior to ingestion, and then 15, 30, 60, 120, and 180 min after ingestion. Urine was obtained for urate and creatinine prior to ingestion, and then 30, 60, 120, and 180 min after ingestion. Urine volume was measured at each time point. Drinking water was also provided at each time point (30, 60, 120, and 180 min) to a volume equivalent to the collected urine volume. Complete study protocol is provided in the Source data file.

**Laboratory testing**. Serum and urine chemistry was tested using the Roche Modular P (Hitachi) analyzer. The FEUA was calculated as the ratio between the renal clearance of uric acid to the renal clearance of creatinine, expressed as a percentage. Urinary excretion (UUE) was also calculated; as the ratio of urinary urate to urinary creatinine. Genotyping of *SLC2A9* rs11942223 and *ABCG2* rs2231142 SNPs was done using TaqMan SNP genotyping assay technology (Applied Biosystems)[30].

**Statistical analysis**. Sample size calculations were based on the variability of change in serum urate at each time point in our previous study of fructose-induced hyperuricemia[46] and of inosine use for Parkinson's disease[44]. We assumed that approximately 30% of participants would have the *SLC2A9* rs11942223 urate-lowering allele, and that approximately 30% of participants would have the *ABCG2* rs2231142 urate-raising allele based on our previous studies[46,47]. The proposed study was adequately powered (90% at the 5% significance level for a two-tailed test) to detect a difference in the change in serum urate between the groups of at least 0.02 mmol/l. Sample size calculations were performed using PASS 2002 (Hintze, J (2006) Kaysville, Utah).

Data are presented as mean (standard deviation (SD)) or $n$ (%) for descriptive purposes; and plotted as least squares adjusted (for age, sex, BMI, and ancestry) mean ± standard error of the mean. The primary endpoint was SU and secondary endpoint was FEUA or UUE. Data were analyzed using a mixed models approach to repeated measures where the dependent variable was modeled in an analysis of covariance (ANCOVA) with age, sex, BMI, and ancestry included as covariates. Secondary analyses included stratifying by ancestry or sex where possible. An unstructured covariance was fitted. Main effects of allele presence/absence, time, and their interaction were constructed. Analyses were performed using SAS (v9.4, SAS Institute Inc, Cary, NC, USA) on an intention to treat basis. All tests were two-tailed. To understand the relationship between SU and FEUA or SU and UUE, we calculated Pearson's coefficient to test if there was a correlation (two-tailed test). We secondarily tested whether SU was dependent on FEUA or UUE using linear regression models with goodness of fit calculations and also compared elevations and slopes of different data sets. To calculate the percentage of the total SU variability that was dependent on FEUA, we calculated the coefficient of determination from the linear regression fit ($r^2$, a percentage of SU variation accounted for by FEUA or renal urate handling) and for the percentage of total SU variability dependent on renal excretion we calculated the coefficient of non-determination ($1-r^2$, percentage of SU variability accounted for by extra-renal function).

**Cell culture and immunostaining in human cells**. Normal human kidney renal epithelial cells from three different adult human kidneys were received from the Baltimore PKD Research and Clinical Core Center (reviewed by the UMB IRB and determined to not be human research, requiring no further IRB review). Primary cell cultures were established by plating cells directly onto Corning Transwells (12 mm tissue culture treated, Costar 3460). Cells were cultured with the Baltimore PKD renal epithelia cell media (REC): 1:1 mixture of RenaLife Complete Medium (Lifeline Cell Technology LL-0025) and Advanced MEM medium (Fisher Scientific #12492) with 5% FBS (ThermoFisher #26140-079), 2.2% Pen/Strep (Fisher Scientific #30-002-Cl), 0.6% L-alanyl-Glutamine (Gemini Bio-products #400-106), and 0.03% Gentamicin (Quality Biological #120-098-661). Human primary cells were cultured for 5 days past confluency to establish a polarized monolayer. On day ten of culture, cells were fixed with 3% paraformaldehyde (Electron Microscope Solutions 15714-S), permeabilized with 0.1% triton (Sigma T-9284), and blocked in 1% BSA (Sigma A6003) as per standard protocols. Following blocking, polarized cells were incubated overnight at 4 ºC with primary antibodies [rabbit anti-ABCG2 (Cell Signaling Technology 42078, 1:200), rabbit anti-URAT1 (MBL BMP064, 1:500), rat anti-zonula occludens 1 (Santa Cruz sc-33725, 1:50), mouse anti-NaKATPase (Millipore 05-369, 1:200)] in 0.1% BSA in 1× PBS. After washing the primary antibody with 1× PBS, the culture was incubated with secondary antibody in 0.1% BSA [1:200 Goat anti-rabbit secondary antibody (Invitrogen A-21429), 1:200 Goat anti-rat secondary antibody (Invitrogen A-21247), 1:200 Goat anti-mouse secondary antibody (Invitrogen A-11001)]. Following this incubation, transwells were washed with 1× PBS and then excised from the plastic mold. The membrane was mounted onto a glass slide with Vectashield Mounting Media with DAPI (Vector Labs H1200). Samples were imaged on a Nikon W-1 Spinning Disk.

**Animal studies**. Animal studies were performed in adherence to the NIH Guide for the Care and Use of Laboratory Animals and approved by the University of Maryland School of Medicine Institutional Animal Care and Use Committee. Mice were housed in groups of two to five per cage on a 12:12 h light/dark cycle; with

lights on at 6 a.m. Tissues, blood, and urine samples were collected between 8 and 10 am. Food and water were available *ad libitum*.

**Metabolic cage studies**. Renal function was evaluated using metabolic cages. After acclimation (2 days) in metabolic cages (Nalgene; Thermo Scientific # NALGE650-0322), food and water consumption was recorded and urine and fecal samples were collected. Urine samples were collected several times a day to prevent contamination from food, water, and fecal matter. Kidney electrolyte handling was assessed in 24-hour measurements of fractional excretion (FE), which are calculated as the rate of urinary excretion of a solute (UxV, where Ux is the urinary concentration of substance, x, and V is the urinary flow rate) relative to the filtered load (FEx = UxV/GFR*Px, where GFR is the glomerular filtration rate as calculated by creatinine clearance (see below), and Px is the plasma concentration).

**Sample collection, preparation, and analysis**. Animals were anesthetized with isoflurane (>4.5%). Once an animal was unconscious, blood samples were collected via cardiac puncture. Blood chemistry and gases (Na+, K+, Cl−, HCO₃−, pH, hematocrit, and BUN) were measured from a 100 μl aliquot of whole blood using an i-STAT EC8+ cartridge and an i-STAT1 Handheld Analyzer (Abaxis). The organs were next harvested (as described below). The remaining fraction of blood was immediately centrifuged ($1000 \times g$) to separate formed elements and plasma, the latter was subsequently isolated and frozen for later analysis of urate, insulin, and IGF1 levels. Urine sodium, potassium, and chloride analysis was performed using an Easylyte Analyzer (Medica Corporation). Plasma and urine creatinine levels were measured using the QuantiChrom Creatinine Assay Kit (BioAssay Systems) following the manufacturer's protocol. Serum urate levels were measured with the HumanSens2.0plus (Human, GER) using UA test strips (#7061). For each set of measurements, a standard curve was calculated using the meter against known urate concentrations, and the measured SU fit to the graph[48]. We independently verified the veracity of the meter at the physiological ranges using two methods: (1) intestinal loop transport studies measuring flux with the Human Sens 2.0 plus meter and using C-14 labeled uric acid and found agreement in measured UA flux; (2) comparison of SU and UA standards using the Uric Acid Assay kit (DIUA-250, BioAssays Systems, USA) and again found agreement. Serum insulin concentrations were measured by rat/mouse ELISA kit according to the manufacturer's instructions (EMD Millipore Corporation, USA, Cat. #EZRMI-13K). Serum IGF-1 concentrations were also measured using a mouse IGF-1 ELISA kit according to the manufacturer's instructions (Sigma-Aldrich, USA, Cat. #RAB0229). Sample measurements were recorded for further calculations using BMG LABTECH's CLARIOstar multi-mode microplate reader.

**Q140K+/+ mouse model creation**. The Q140K+/+ mice were generated using CRISPR/Cas9 on a C57BL6J (#000664) background by JAX laboratories. The knock-in (KI) was accomplished using sgRNA guide #2 GTAAGTTTTCTCTC ACTGTC, chosen because of the low predicted off target score (33.8). Eighty three F0 mice were generated after guide injection, and 4 founders (2 males, 2 females) identified without deletions in the *Abcg2* loci, and bred with WT C57BL6J mice to produce 31 N1 generation mice. Of these, a subset was identified from a single founder that possessed the KI, but no other deletions in the *Abcg2* gene. The heterozygotic offspring from the single founder x WT C57BL6J pair were then used to generate two inbred lines; one a Q140K+/+ line and one Q140K−/− (WT) line. Experimental animals used for comparisons (metabolic cage experiments) all shared Q140K+/− grandsire and grandam. This breeding strategy was employed to ensure that off target effects, if any, resulting from nonspecific guide binding would be constant across all mice tested. Genotyping was accomplished with the primers: fw: GGT GTA CGG CTG GGT AAT GA; rv: AAA GCT GCC AAA GGA AAG AC.

**Immunostaining fixed kidney tissues**. Mice were euthanized and kidneys dissected. Normal human cortex tissue was dissected from nephrectomized human kidney tissue received from the Baltimore PKD Research and Clinical Core Center. All renal tissue samples were washed in 1× PBS and rocked overnight at 4 °C in 3% paraformaldehyde (PFA). Following fixation, tissue samples were washed three times with 1× PBS for 10 min each and stored until embedding in 70% ethanol at 4 °C. Tissue samples were then embedded in paraffin, sectioned, and mounted onto glass coverslips with a gelatin coating solution (50 ml distilled water, 0.25 g Gelatin, 25 mg Chrome Alum). Deparaffination of tissue samples was performed by washing coverslips with xylene and ethanol (5 min in 100% Xylene, 5 min in 100% Xylene, 5 min in 100% Ethanol, 5 min in 95% Ethanol, 5 min in 70% Ethanol, 5 min in distilled water, 5 min in distilled water). Following the last wash, cover slips were placed in a heat-induced epitope retrieval (HIER) solution, pH 8.0 (1 mM Tris (American Bioanalytical AB02000-01000), 0.5 mM EDTA (Sigma E5134)) in final volume of 300 mL of distilled water with 0.02% SDS (American Bioanalytical AB01920-00500). Samples were warmed in HIER solution with SDS to 100 °C, then transferred to a 100 °C water bath for 15 min. Following this incubation, samples were allowed to cool to room temperature, and washed with distilled water and 1× PBS. Samples were treated with 2–3 drops of Image-iT FX Signal Enhancer (Molecular Probes 136933) for 15 min with rotation at room temperature, and then blocked in Incubation Media [1% BSA (Sigma A7638), 0.1%

Tween 20 (BioRad 170-6531), 0.02% sodium azide (Sigma S2002) in a final volume of 50 mL 1× PBS (Bio-Rad #161-0780)] with 1% donkey serum (Sigma D9663) for another 15 min with rotation at room temperature. Primary antibody was added in incubation media and serum solution [rabbit anti-ABCG2 (Cell Signaling Technology 42078, 1:200), rabbit anti-URAT1 (MBL BMP064, 1:500, biotinylated LTL (Vector Labs B1325, 1:100), mouse anti-NaKATPase (Millipore 05-369, 1:200)] and incubated overnight in a humidifier chamber at room temperature. Coverslips were then quickly washed three times in 1× PBS, with a fourth wash lasting for 30 min. Secondary antibodies [1:200 Goat anti-rabbit secondary antibody 555 (Invitrogen A-21429), 1:50 Streptavidin conjugated 647 (Invitrogen 43-4316), 1:200 Goat anti-mouse secondary antibody (Invitrogen A-11001)] were added in incubation media with donkey serum and incubated at room temperature, in the dark for two hours. Coverslips were then again quickly washed (3×) in 1× PBS with a fourth wash lasting 30 min. Following washes, coverslips were mounted onto glass slides with Vectashield Mounting Media [Vector Labs H1000], and sealed with nail polish. Samples were imaged on an Olympus IX83 inverted imaging system. To quantify signal intensities images were acquired on Olympus I × 83 with matched exposure times between mutant and control experimental groups per experiment in MetaMorph (Molecular Devices, CA). Additionally, each image was scale acquisition (contrast corrected) controlled between mutant and control groups per experiment in MetaMorph. Images were then thresholded by subtracting background from the mutant group and converted to an RGB image. Then, the Color Pixel Counter Plugin in FIJI (open source platform based on Image J) was used to count the number of positive pixels above the threshold. Changes in the ratio of positive thresholded pixels to total pixels (PTP/TP) is reported.

**Immunostaining fixed intestinal and liver tissues**. For staining of mouse intestinal tissue sections, the intestine was dissected, and the ileum was rinsed with ice-cold saline and fixed in 10% formalin before paraffin embedding[49]. Individual sections were heat fixed and deparaffinized followed by microwave treatment in 0.01 M sodium citrate solution (pH 6.0) for antigen recovery. Sections were then washed in 1× PBS and blocked with 5% normal goat serum in PBS for 1 h at room temperature. Subsequently, sections were incubated with primary antibody diluted (1:100) in 5% normal goat serum in PBS overnight at 4 °C followed by FITC- or Alexa-conjugated goat secondary antibody (1:200), and images were obtained using a Zeiss LSM 510 confocal microscope. Liver samples were dissected and either flash frozen or fixed in 3% PFA before embedding and staining with hematoxylin and eosin.

**Western blot protocol**. Freshly dissected kidney tissue was homogenized using the BeadBug™6 (Benchtop Scientific) using 3.0 mm beads (Benchmark Scientific, #D1032-30) with a RIPA lysis buffer (1% deoxycholic acid, 1% triton X-100, 0.1% SDS, 150 mM NaCl, 1 mM EDTA, 10 mM Tris HCl pH 7.5 and 1:100 protease inhibitor (complete protease inhibitor, Sigma, USA)). The samples were incubated at 37 °C for 30 min with 5× laemmli buffer and 5% 2-beta mercaptoethanol. The samples were run on precast 10% Stain-Free Gels (Biorad, USA), transferred to nitrocellulose membrane using the Bio-Rad Transblot transfer system (Biorad, USA), and blocked in 5% nonfat milk. The appropriate primary antibody (anti-SLC22A12; MBL International Corp (JPN), #BMP064, 1:1000; rabbit anti-ABCG2 (Cell Signaling Technology 42078, 1:1000) was added and the membrane was incubated overnight at 4 °C. After washing and the addition of appropriate secondary antibody, the membrane was exposed with Super Signal ECL (Pierce), chemiluminescence signal was captured using a ChemiDoc system (BioRad, USA), and the band density was calculated using BioRad's Image Lab Software. Normalization was done by calculating the total protein loaded in each lane using the BioRad Stain-Free Gel System. Statistical comparisons of density measurements from western blots were done with the Student's *t*-test for pair-wise comparisons, or an ANOVA, used with a Tukey's or Dunnett's Test for multiple comparisons (Prism 7, GraphPad, USA). All reported means are ± standard error of the mean (SEM). For intestinal tissues, samples from mouse small intestinal tissue were washed and scraped with a cover glass in PBS as described[50]. Crude extracts were prepared by sonication in lysis buffer containing complete protease inhibitor (1:100) followed by brief sonication on ice. The samples were then treated as described above for kidney tissue.

**Quantitative real-Time PCR (qRT-PCR)**. Whole kidney (or intestine or liver) preserved in RNAlater™ (Qiagen, #1017980) was homogenized by bead beating using a BeadBug™6 (Benchtop Scientific) using 3.0 mm beads (Benchmark Scientific, #D1032-30). RNA was isolated using TRIzol™ (Ambion, #15596026) using Phase Lock Gel Heavy tubes (Quantabio, #2302830) for phase separation followed by RNA clean up using the RNeasy® Mini Plus kit (Qiagen, #74134). 5 μg of RNA was then transcribed into cDNA using the SuperScript™ III First-Strand Synthesis System (Invitrogen, #18080051). 10 ng of the resulting cDNA was used as template for qRT-PCR using PowerUp™ SYBR™ Green Master Mix (Applied Biosystems, #A25742), run using the QuantStudio 3 Real Time PCR System (Applied Biosystems). Manufacturer's specifications were followed for all reagents listed above. Primers were validated previously[51–57] and used the following sequences: (ABCG2: F-AAACTTGCTCGGGAACCCTC, R-CTCCAGCTCTATTTTGCATTCC,; GAPDH: F-CTTTGGCATTGTGGAAGGGC, R-TGCAGGGATGATGTTCTG

GG; Fasn: F-GCTGCGGAAACTTCAGGAAAT, R–AGAGACGTGTCACTCCT GGACTT; PNPLA2: F-TATCCGGTGGATGAAAGAGC, R-CAGTTCCACCTGC TCAGACA; G6PD: F-CCGGAAACTGGCTGTGCGCT, R–CCAGGTCACCCGA TGCACCC, PNPLA3: F-CGAGGCGAGCGGTACGT, R–TGACACCGTGATGG TGGTTT; SREBP-1a: F-GCGCCATGGACGAGCTG, R–TTGGCACCTGGGCT GCT; SREBP-1c: F-GGAGCCATGGATTGCACATT, AGGAAGGCTTCCAGA GAGGA).

**Uric acid transport study in ex vivo intestinal loop**. Following anesthesia by isoflurane, a laparotomy was performed, and experimental loops of 3–5 cm length were constricted at the small intestine by tying with nonabsorbable silk. For a typical experiment, 4–6 loops were isolated from the small intestine and placed into 100 mm × 20 mm dish containing prewarmed (37 °C) Ringer's (in mM: 140 NaCl, 5 KCl, 1 MgCl$_2$, 2 CaCl$_2$, 10 HEPES, 10 glucose, pH 7.4) of specific urate concentration, then Ringer's (100–200 μl) was injected in each loop via a tuberculin syringe through the ligated end of the loop to allow for transport of urate from serosal (s) to mucosal (m). After 10 mins, these loops were placed in ice cold Ringer's (no urate). The fluid from each loop was collected, and the amount of urate in the loop with respect to the surface area of the loop (μM/cm$^2$/time) was calculated. For mucosal to serosal urate transport, loops were filled with 1 mM urate and submerged into an Eppendorf tube containing 150–200 μl Ringer's solution at 37 ºC. Unidirectional fluxes were directly measured and used to calculate the overall net direction of urate movement (J$^{UA}$net) across the epithelium by the formula (J$^{UA}$net = J$^{UA}$ms – J$^{UA}$sm)[58]. ABCG2 inhibitor, topiroxostat (FYX-051, cat no. HY-14874), was purchased from MedChemExpress USA.

**Statistical analysis of mouse and cell work**. Data were subjected to t-tests (for paired or unpaired samples as appropriate) and ANOVA for statistical analysis and presented as means ± SEM. Statistical analyses were carried out using Prism 7 software. $P \leq 0.05$ was considered significant. Specific details on each statistical test can be found in the figure legends or text of the "Results" section.

## Data availability
Source data underlying Figs. 1–7, Table 1, Supplementary Figs. 1–7, and Supplementary Table 2 are provided as a Source data file. The human interventional trial was registered on the Australian New Zealand Clinical Trials Registry (https://www.anzctr.org.au/Trial/Registration/TrialReview.aspx?id=369688), number ACTRN12615001302549. The full protocol can be found in the Source data file. Additional data will be provided by the corresponding author upon reasonable request.

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

## Acknowledgements

The authors gratefully acknowledge the Baltimore PKD Research and Clinical Core Center for providing normal human kidney cells and tissues. The work of O.M.W. was supported by NIDDK grants R01DK114091 and 5P30DK090868. The views expressed in this paper are those of the authors and do not necessarily represent the views of the National Institutes of Health. N.D., T.R.M., L.K.S., and M.P.G. acknowledge the support of the New Zealand Health Research Council via Program grant 14/527.

## Author contributions

K.M.H.: concept/design; acquisition; analysis; interpretation of data. E.E.D.: concept/design; acquisition; analysis; interpretation of data. R.M.L.: acquisition. J.A.: acquisition. G.D.G.: analysis. A.P.: acquisition. V.L.H.K.: acquisition. A.H.: acquisition. L.K.S.: concept/design; analysis; interpretation of data. T.R M.: concept/design; acquisition; analysis; interpretation of data; substantively revised paper. N.D.: concept/design; acquisition; analysis; interpretation of data; substantively revised paper. O.M.W.: concept/design; acquisition; analysis; interpretation of data; drafted paper; substantively revised paper.

## Competing interests

In the last 3 years, N.D. has received speaking fees from Pfizer, Horizon, Janssen, and AbbVie; consulting fees from Horizon, Hengrui, and AstraZeneca; and research funding from Amgen and AstraZeneca. No other authors declare any competing interests.
