## [Peer Review File · Nature Communications]

Reviewers' Comments:

Reviewer #1:

Remarks to the Author:

This is an interesting paper that documents an effect of the ABCG2 variant Q141K (the 141 risk allele) in causing hyperuricemia via impaired intestinal excretion. The finding has already been suggested but the approach provided add a lot of important aspects and I commend the authors for their efforts. I do have some concerns

1. The effect of the variant on renal excretion was hard to follow. One might expect that blocking intestinal secretion would lead to hyperuricemia and elevated FEUA, but it seems that the higher the serum uric acid, the lower the FEUA suggesting some lack of renal compensation. I definitely had trouble understanding this and wonder if it might be explained more simply. Perhaps a figure showing the proposed normal handling and handling when the variant is expressed, both in the normal condition and following a purine load.
2. One major finding that seems very key to the discussion is the difference between males and females, which was noted in both the animal and human studies, but the results as presented rarely discuss the differences in experimental findings. It seems that the authors likely have the data but need to discuss differences in localization and function in their studies.
3. Another interesting finding is that the mouse model is associated with hyperinsulinemia and elevated glucose, that seems most consistent with insulin resistance. One might posit that impaired secretion in the intestine would lead to high delivery to the liver, and it would be of interest to know if the animals develop fatty liver as well. Some comment on the female animals is important as well. It seems like the male animals may be developing features of metabolic syndrome.
4. The human data included inosine loading but the numbers of individuals with the variants and sex are not mentioned, and one might expect differences from the male and female. I also wonder if inosine loading in the mouse model is key. One thought i have is that the alteration in ABCG2 might lead to differences in uric acid absorption and or production, and that it might not just be an intestinal-renal cross talk.

Reviewer #2:

Remarks to the Author:

This manuscript addresses the urate handling influenced by the common ABCG2 variant, Q141K using human interventional study and Q140K Abcg2 variant mice. The authors examined the mechanism in detail and demonstrated reduction of extra-renal urate excretion in the ABCG2 variant, resulting in hyperuricemia. Clarification of the mechanism is important for the prevention of gout. However, the findings include some confirmatory findings for prior works.

The authors performed the loading test for Q141 and Q141K participants. The results were increase of SUA and no change of FEUA in Q141K participants, comparing to Q141 participants, though FEUA was related with SUA at 180 minutes.

How did the urinary urate excretion change in the loading test? Urinary urate excretion should be shown.

Fig 1-E and 1-F are interesting and novel, as measurement of intestinal urate excretion has not been reported. Unfortunately, the calculation method of %extra renal contribution to SUA is not described in the methods section. Thus, we could not evaluate adequacy of the method and these results. Does extra renal contribution exceeding renal urate contribution indicate that extra renal excretion is larger than renal excretion? If not, what kind of status is conceivable? If it indicate that, this result should be emphasized more.

Purine metabolism and renal environment are different between human and mice. Mice have

adapted to different circumstance that the final metabolite of purine is not uric acid. For example, renal failure occurs in uricase knockout mice. Thus, it is difficult to evaluate hyperuricemic mouse model though findings in drastic environments such as knockout mice are helpful. Fortunately, the results in the knock-in mice were similar to that in human. Some of the findings are similar to Abcg2 knockout mice in the previous report.

Other minor points

In method section, eGFR is not calculated from creatinine clearance.

In the results, Q141 should be Q141K in line 19, pp 7.

Urinary urate excretions were not significant between Q140K mice and wild type mice. But it looks like urinary urate excretions in Q140K will significantly increase comparing wild type mice, if the number of mice increases.

In method section, the acquisition way of topiroxostat should be written.

Reviewers' comments:

Reviewer #1 (Remarks to the Author):

This is an interesting paper that documents an effect of the ABCG2 variant Q141K (the 141 risk allele) in causing hyperuricemia via impaired intestinal excretion. The finding has already been suggested but the approach provided add a lot of important aspects and I commend the authors for their efforts. I do have some concerns

1. The effect of the variant on renal excretion was hard to follow. One might expect that blocking intestinal secretion would lead to hyperuricemia and elevated FEUA, but it seems that the higher the serum uric acid, the lower the FEUA suggesting some lack of renal compensation.

We apologize for the lack of clarity in our discussion of the effect of the 141K/Q140K variant on renal function. We have altered the language in both the results and discussion to make it easier to understand. The reviewers' assumptions of what should occur are correct. In the human cohort, as the SU increased so did the FEUA (Figure 1A and Figure 1B), this was true in both Q141 and 141K individuals. What was surprising was that as the SU increased its dependence on FEUA actually decreased. In other words, at higher SU levels the renal handling of urate becomes less important for determining the final SU levels. This appeared to be because there was a cap to how high FEUA and renal excretion can go, at levels exceeding this cap the extra-renal secretion (mainly the intestines) handled the extra UA. However, the 141K individuals appeared to largely lack this excretion reserve. We have now summarized these ideas in a summary figure in the discussion section (Figure 8).

I definitely had trouble understanding this and wonder if it might be explained more simply. Perhaps a figure showing the proposed normal handling and handling when the variant is expressed, both in the normal condition and following a purine load.

We agree and have added a new summary figure, Figure 8, in the discussion section.

2. One major finding that seems very key to the discussion is the difference between males and females, which was noted in both the animal and human studies, but the results as presented

rarely discuss the differences in experimental findings. It seems that the authors likely have the data but need to discuss differences in localization and function in their studies.

We agree that differences in sex and how urate is handled is of great importance. In our human cohort there were too few males, especially with the 141K variant, to allow for a statistically meaningful analysis. However it is important to note that our analysis of the human data did correct for a number of variables, including sex (please see methods section). The mice, however, do offer an opportunity for us to include further data. We have concluded a second set of metabolic cage experiments comparing females of both WT and Q140K+/+ genotypes and now include these data in a new supplemental figure (Supplemental Figure 3) and a new supplemental table (Supplemental Table 2). We found the female Q140K+/+ mice were, in large part phenotypically normal, they had normal SUA, normal FEUA, and largely normal blood and urine chemistries. Even if these data are nominally negative results, they provide significant support to the hypothesis that the Q141K variant preferentially is pathogenic in males and that there is either a genetic or physiological protection in females that preserves the function of Q141K ABCG2. We have added these ideas to our discussion section (end of page 17 and beginning of page 18). The phenotypic differences found here with the Q140K+/+ mice echo what has been shown true in large human populations, with the Q141K variant having a stronger effect on serum urate in males.

3. Another interesting finding is that the mouse model is associated with hyperinsulinemia and elevated glucose, that seems most consistent with insulin resistance. One might posit that impaired secretion in the intestine would lead to high delivery to the liver, and it would be of interest to know if the animals develop fatty liver as well. Some comment on the female animals is important as well. It seems like the male animals may be developing features of metabolic syndrome.

We agree with the reviewer that potential metabolic changes and structural changes to the liver tissue are of great interest. We inspected livers from 6 male WT and 6 male Q140K animals and did observe sporadic fatty liver phenotypes, but only in the Q140K animals. In a new supplemental figure, Supplemental Figure 4, we present images from all 12 livers to demonstrate the spectrum of phenotypes, from none to significant. We also investigated gene expression levels of a number of markers for fatty liver disease [*Srebp-1a*, *Srebp-1c*, *Pnpla2*, *Pnpla3*, *Fasn*, and *G6pd*] in those animals that had fatty liver phenotypes and compared them to their matched controls and found three of the genes (*Pnpla2*, *Pnpla3*, and *G6pd*) showed alterations in mRNA levels (data presented in a new primary figure, Figure 4, and Supplementary Figure 4). In the female mice we saw an interesting alternative narrative. The Q140K+/+ females were not hyperuricemic, and they had no alterations in glucose, insulin, or IGF1, data we have now added to Supplemental Figure 3 and Supplemental Table 2. We have also added a section in the discussion (bottom of page 18) to address these observations and to comment on the significance of the apparent alteration in metabolism in the Q140K animals. We greatly appreciate the reviewers' comments and suggestions, these data have improved the manuscript.

4. The human data included inosine loading but the numbers of individuals with the variants and sex are not mentioned, and one might expect differences from the male and female. I also wonder if inosine loading in the mouse model is key. One thought I have is that the alteration in ABCG2 might lead to differences in uric acid absorption and or production, and that it might not just be an intestinal-renal cross talk.

We appreciate the reviewers' interest in the population statistics of our human cohort. We have summarized the characteristic of the population in Supplementary Table 1. As noted earlier the discrepancy in numbers of each sex, especially with respect to the Q141K variant, prevented a sex-stratified analysis.

In the work presented in the manuscript we did not load the animals with inosine to drive the hyperuricemia, as we did in the humans, what we observe and report in mice is spontaneous and baseline hyperuricemia. Thus, the alterations to ABCG2 cannot contribute to altered inosine reabsorption. In the human population we observe a similar rate of increase of SU over time for both genotypes, superficially suggesting that the rate of inosine absorption and conversion to UA is similar. However because we did not perform inosine loading in the mice we cannot rule out that loss of ABCG2 function in the mouse liver could change acute liver UA levels altering the efficiency of either UA production, or metabolism. This is an interesting question, but beyond the scope of the current manuscript. However as discussed above we did observe other alterations in liver architecture and function and these may contribute to the hyperuricemia. Therefore, we have added a section in the discussion addressing potentials effects the Q140K mutation and or hyperuricemia has on liver function and metabolism (bottom of page 18).

Reviewer #2 (Remarks to the Author):

This manuscript addresses the urate handling influenced by the common ABCG2 variant, Q141K using human interventional study and Q140K *Abcg2* variant mice. The authors examined the mechanism in detail and demonstrated reduction of extra-renal urate excretion in the ABCG2 variant, resulting in hyperuricemia. Clarification of the mechanism is important for the prevention of gout. However, the findings include some confirmatory findings for prior works. The authors performed the loading test for Q141 and Q141K participants. The results were increase of SUA and no change of FEUA in Q141K participants, comparing to Q141 participants, though FEUA was related with SUA at 180 minutes.

1. How did the urinary urate excretion change in the loading test? Urinary urate excretion should be shown.

We appreciate the reviewers' suggestion that inclusion of the urinary urate excretion (UUE: U_{ua}/U_{cr}) would support the other work presented here and thus we have added the UUE for the Q141 and Q141K variant individuals during the inosine challenge to Figure 1 (Figure 1C). The UUE, as expected increases significantly over time, however there is no significant differences between the Q141 and 141K individuals. We also added a comparison of the SU and its dependence on UUE at baseline (before loading, Figure 1H) and have found that the 141K individuals show a significant correlation but Q141 individuals do not. These two new sets of data further confirm the loss of extrarenal excretion (i.e. the importance of renal excretion). The UUE data does not however demonstrate support for the over-load hypothesis that has been proposed by others. There are a number of possible explanations for these differences and we have discussed some in a new addition to the discussion section (Page 15, 2nd paragraph).

2. Fig 1-E and 1-F are interesting and novel, as measurement of intestinal urate excretion has not been reported. Unfortunately, the calculation method of %extra renal contribution to SUA is not described in the methods section. Thus, we could not evaluate adequacy of the method and these results.

We apologize for the lack of clarity. We have expanded the method section where we discussed our calculation methods, this is an important calculation and we appreciate the reviewer

expressing the need for more details. We would like to also reiterate that we are not measuring extra-renal excretion *per se* in the human cohort, what we have done is to model the contribution of the extra-renal excretion on SU levels. In short, we have statistically evaluated the dependence of SU on FEUA (renal urate handling) and found only partial dependence (R^2 is not 1) and calculated what contribution everything but FEUA (ie extra renal) makes on SU levels. The critical finding was that in the 141K individuals SU is much more dependent on FEUA, suggesting a loss of the extra-renal contribution. We then tested this hypothesis in the mouse model and confirm that the Q140K mice have significantly reduced intestinal excretion, but preserved kidney function, much like what we found in the 141K humans.

3. Does extra renal contribution exceeding renal urate contribution indicate that extra renal excretion is larger than renal excretion? If not, what kind of status is conceivable? If it indicate that, this result should be emphasized more.

We have not measured extra-renal excretion in the human cohort. We have modeled the contribution the extra-renal cohort makes on SU as compared to renal urate handling (FEUA), and we find the extra-renal contribution to setting the SU levels is diminished in the Q141K individuals. From our data it is hard to determine absolute extra-renal excretion amounts. We speculate that much of the variability of SU between individuals and in response to a urate load may be due to variability in extra-renal excretion. We propose a model where kidney urate handling is static, and the extra-renal excretion pathway provides the further capacity needed with high urate loads, a model we now summarize in a new figure, Figure 8, in our discussion section. In this model, increased urate load increases FEUA and UUE because the reabsorption and active secretion pathways are already at capacity at baseline but soon reach a maximum or cap, whereas the extra-renal excretion may play a large role in handling any additional UA in the system.

4. Purine metabolism and renal environment are different between human and mice. Mice have adapted to different circumstance that the final metabolite of purine is not uric acid. For example, renal failure occurs in uricase knockout mice. Thus, it is difficult to evaluate hyperuricemic mouse model though findings in drastic environments such as knockout mice are helpful. Fortunately, the results in the knock-in mice were similar to that in human. Some of the findings are similar to *Abcg2* knockout mice in the previous report.

We agree that using mice for modeling human urate handling has many challenges, some of which we encountered (e.g. lack of SLC2A9 in the mouse proximal tubule). Hence our use of a more human-relevant Q140K model, rather than the more extreme knockout. However, as we report here, we also found that there are many similarities between humans and mice in their UA handling and significant insights are achievable. We appreciate the reviewers' comments and have been careful not to over interpret the model in our discussion section.

5. Other minor points

In method section, eGFR is not calculated from creatinine clearance.

We appreciate the reviewer's discovery of our oversight and have altered the text to reflect that that eGFR and GFR, as determined by creatinine clearance in the mice, are not calculated in the same manner, and we no longer refer to the GFR calculations in mice as eGFR.

6. In the results, Q141 should be Q141K in line 19, pp 7.

Yes it should be, we have changed it, and greatly appreciate the diligence of the reviewer.

7. Urinary urate excretions were not significant between Q140K mice and wild type mice. But it looks like urinary urate excretions in Q140K will significantly increase comparing wild type mice, if the number of mice increases.

The inherent variability in working with physiological measurements in mice often results in all but the most robust changes falling short of significance. Before performing the reported metabolic cage experiments for measuring SU, FEUA, and UUE, we used pilot data to do a power calculation to determine how many animals would be necessary to achieve sufficient statistical power, and even with sufficient power, there was no significant alterations in UUE. However we agree with the reviewer that the variance in the UUE data is large, and therefore we cannot rule out the possibility that significantly more N's could alter our conclusions. We have thus have added a sentence in the results noting the large variance and softened the language of the conclusions drawn from these data.

8. In method section, the acquisition way of topiroxostat should be written.

We appreciate the reviewers catching our oversight and we have added this information to the methods section.

Reviewers' Comments:

Reviewer #1:

Remarks to the Author:

We thank the authors for addressing our suggestions. The paper is improved and will make a nice contribution to the field

Reviewer #2:

Remarks to the Author:

This manuscript was amended properly, according to the comment. This study demonstrates how ABCG2 Q141K works in normouricemic individuals well. Renal overload, one of mechanisms for hyperuricemia, was realized by oral inosine load. The different results of FEUA and UUE from that of hyperuricemic patients previously reported might come from genetic differences between normouricemic individuals and hyperuricemic individuals. It would be better to add this point in the manuscript.

Reviewer #3:

Remarks to the Author:

Mode of inheritance for rs2231142:

What is the evidence that the mode of inheritance for the ABCG2 SNP rs2231142 is dominant and not additive? Or what is the reason to show carriers vs non-carriers? I would imagine there would be more power for the additive model and that showing the dose response effects of the number of 141K alleles (vs carriers/non-carriers) could be stronger evidence in support of the author's conclusions. Is it difficult to incorporate the additive effect in the mixed models that were used?

Population stratification and confounding:

I'm a bit concerned about population stratification since only ancestry is included in the statistical model. Since the authors do not have genome-wide data and cannot adjust for measures of population stratification, could the authors present the results stratified by ancestry in the supplemental materials to demonstrate that the marginal effects are similar across all ancestries and that combining all ancestries in one analysis does not violate any assumptions of the statistical model? (i.e. Calculate the minor allele frequencies of the risk inducing allele in each of the ancestries to show they are similar. Show there are no differences in SU, FEUA or UUE by ancestry. Show associations of rs2231142 with SU, FEUA and UUE stratified by ancestry and that there are no qualitative differences by ancestry.) I think this would show the data and the analyses are robust.

Sex-stratified analysis:

I agree with the authors that there is limited power to perform the analysis among the male participants in the interventional study. Perhaps a potential way around this limitation could be if the authors presented the association of rs2231142 with SU among the female participants to see if the marginal effects between rs2231142 and SU from the combined sex analysis are driven by the male participants? Does this indicate that a SNP*sex interaction term needs to be introduced into the model?

While the effect estimates for rs2231142 and SU are statistically different in the cited papers, the associations are not qualitatively different (effect estimates are not in opposing directions). It does appear the effect of rs2231142 in males is stronger than in females. Is this clinically significant and can the assumption be made that the 141K/140K variant is fully functional in females?

Lines 194-196 – or if there is an effect it is not larger than that of rs1194222331 in SLC2A9 (under

the assumption that this effect size is clinically meaningful) and therefore this study did not have enough power to detect it.

Lines 394-398 – If there are genetic differences by ancestry, the effect could be masked by combining different ancestries in the same analysis. The authors can address this by showing the results of the analysis stratified by ancestry as detailed in my second set of comments.

Reviewers' comments:

Reviewer #1 (Remarks to the Author):

We thank the authors for addressing our suggestions. The paper is improved and will make a nice contribution to the field

We appreciate the extremely helpful previous comments from Reviewer #1 and believe their efforts have increased the value of this manuscript to the field.

Reviewer #2 (Remarks to the Author):

This manuscript was amended properly, according to the comment. This study demonstrates how ABCG2 Q141K works in normouricemic individuals well. Renal overload, one of mechanisms for hyperuricemia, was realized by oral inosine load. The different results of FEUA and UUE from that of hyperuricemic patients previously reported might come from genetic differences between normouricemic individuals and hyperuricemic individuals. It would be better to add this point in the manuscript.

We appreciate the insights Reviewer #2 has provided and we agree that our study provides interesting contrasts with previous studies. There are numerous explanations for the differences in findings and we have included in our discussion a point that the ancestry of the cohorts are different and that this may help explain differences, with secondary common variants in other genes contributing (previous studies were based on male Asian cohorts, here we use non-Asian populations). Also possible is the difference in the sex make-up of the two cohorts. Previous studies have focused on male only cohorts, here we used a mixed population. An analysis of the males and females (non-carriers of risk allele) show differences in kidney function of males and females, including the UUE as the urate load increases (Supplementary Figure 7). This sex difference may contribute to our lack of resolving the renal overload in the Q141K carriers. However, as renal overload is an emerging and interesting hypothesis to explain some forms of hyperuricemia, we have added to our discussion language to incorporate the Reviewers' comments about differences between normouricemic cohorts, as well as the possible sex dependence (page 15 and 16).

Reviewer #3 (Remarks to the Author):

Mode of inheritance for rs2231142:

What is the evidence that the mode of inheritance for the ABCG2 SNP rs2231142 is dominant and not additive? Or what is the reason to show carriers vs non-carriers? I would imagine there would be more power for the additive model and that showing the dose response effects of the number of 141K alleles (vs carriers/non-carriers) could be stronger evidence in support of the author's conclusions. Is it difficult to incorporate the additive effect in the mixed models that were used?

The Reviewer raises an interesting question and we agree with the increased power of an additive model. Unfortunately, our study only contained a single individual with two copies of the Q141K variant allele, preventing an additive model analysis.

Population stratification and confounding:

I'm a bit concerned about population stratification since only ancestry is included in the statistical model. Since the authors do not have genome-wide data and cannot adjust for measures of population stratification, could the authors present the results stratified by ancestry in the supplemental materials to demonstrate that the marginal effects are similar across all ancestries and that combining all ancestries in one analysis does not violate any assumptions of the statistical model? (i.e. Calculate the minor allele frequencies of the risk inducing allele in each of the ancestries to show they are similar. Show there are no differences in SU, FEUA or UUE by ancestry. Show associations of rs2231142 with SU, FEUA and UUE stratified by ancestry and that there are no qualitative differences by ancestry.) I think this would show the data and the analyses are robust.

We thank the Reviewer #3 for the suggestion. We have rerun our analysis stratified by ancestry and have included this data in Supplemental Figures 1 and 2 and Supplemental Table 1. The minor allele frequencies are similar (0.09 and 0.13) among the two populations, but there is no qualitative differences in FEUA or UUE. The SU in the European ancestry group is not significantly different in carriers of the Q141K allele. However, the low power (there are only 8 Q141K individuals of European descent) means that with the assumption that 70% of one standard deviation is physiologically relevant, then our power to detect that difference is only 43% (80% is desirable). Thus, interpretation of this discrepancy is difficult. Overall, the results stratified by ancestry largely support our findings and that our analysis is robust.

Sex-stratified analysis:

I agree with the authors that there is limited power to perform the analysis among the male participants in the interventional study. Perhaps a potential way around this limitation could be if the authors presented the association of rs2231142 with SU among the female participants to see if the marginal effects between rs2231142 and SU from the combined sex analysis are driven by the male participants? Does this indicate that a SNP*sex interaction term needs to be introduced into the model?

We agree that the Reviewer raises an interesting question. We have done an all-female analysis (Supplemental Figure 7) and an analysis of the data with a SNP*sex interaction term (summarized in Supplemental Table 3) and found no differences in the results, supporting our original analysis and arguing against the inclusion of a SNP*sex interaction term in the model.

While the effect estimates for rs2231142 and SU are statistically different in the cited papers, the associations are not qualitatively different (effect estimates are not in opposing directions). It does appear the effect of rs2231142 in males is stronger than in females. Is this clinically significant and can the assumption be made that the 141K/140K variant is fully functional in females?

The Reviewer raises important and complicated questions, questions that are difficult to resolve using human clinical data. This is one of the benefits of our mouse model to study these questions. We show that the mutant protein in the female mice appears to be regulated differently than in the males, specifically the abundance of the mutant protein is significantly higher than in the male mice. This finding is consistent with the differences in FEUA and consistent with increased ABCG2 function in the females. Is the female Q141K ABCG2 fully functional? Probably not, however the increased abundance does appear to alter the overall kidney handling of urate in females (thus FEUA similar to the Wt female mice).

Lines 194-196 – or if there is an effect it is not larger than that of rs11942223 in SLC2A9 (under the assumption that this effect size is clinically meaningful) and therefore this study did not have enough power to detect it.

This is an important point and we have added language acknowledging the possible underpowered study explanation to the top of page 8.

Lines 394-398 – If there are genetic differences by ancestry, the effect could be masked by combining different ancestries in the same analysis. The authors can address this by showing the results of the analysis stratified by ancestry as detailed in my second set of comments.

We appreciate the thoughts of the Reviewer and as requested in a previous comment, we have run an analysis stratified by ancestry with results very similar to those of the inclusive model. It is important to note that in the context of these specific lines in the discussion, the ancestry difference refers to the results in the published literature, based on Asian male cohorts. Our participants do not include people of Asian ancestry, thus in this specific case the analysis stratified by ancestry does not help address the apparent differences observed between our findings and those published earlier. However, we have addressed in our response to Reviewer #2 (above) the differences in uricemic state and in sex make-up of the cohorts as potential important explanations, and we have added this to our discussion (pages 15 and 16).

Reviewers' Comments:

Reviewer #3:

Remarks to the Author:

My concerns have been adequately addressed. No further comments.

REVIEWERS' COMMENTS:

Reviewer #3 (Remarks to the Author):

My concerns have been adequately addressed. No further comments.

Thank you for your contribution to the work.